# Towards Faster and Stronger Deep Earth Mover's Distance for Few-Shot Learning

## Abstract

Recent works in few-shot learning (FSL) for visual recognition have indicated that dense features benefit representation learning across novel categories. One of particularly interesting methods is DeepEMD that is formalized as optimal matching of dense features via an effective statistical distance, i.e., Earth Mover's Distance. Despite its competitive performance, DeepEMD is computationally very expensive due to inherent linear programming. Towards addressing this problem, we propose a metric-based Gaussian EMD (GEMD-M) for FSL. We adopt Gaussians for modeling distributions and closed form EMD between Gaussians as a dissimilarity measure. We illuminate that this metric amounts to feature matching, in which the optimal matching flows follow a joint Gaussian and can be expressed analytically. As the distance in GEMD-M is entangled and not that GPU-friendly, we further present a transfer learning-based Gaussian EMD (GEMD-T). The key idea is to learn a parametric EMD for a more discriminative metric based on square-roots of covariance matrices (via learnable orthogonal matrices) and mean vectors. The learnable metric in GEMD-T is decoupled and thus can be implemented by a fully-connected layer followed by a softmax classifier, very suitable for GPU. We conduct extensive experiments on large-scale Meta-Dataset and three small-scale benchmarks. The results show our GEMD is superior to DeepEMD and achieves compelling performance compared to state-of-the-art methods.

## 1 Introduction

Few-shot learning (FSL) for visual recognition aims to classify novel categories unseen previously given only a limited training examples (Fei-Fei et al., 2006; Koch et al., 2015; Finn et al., 2017). Typically, FSL is formulated as a multitude of episodes (Vinyals et al., 2016), for each of which one trains a network model to classify images of a query set, given a support set containing novel classes each with few training images (shots). Despite great advance in deep learning, it is still challenging to learn novel knowledge with scarce training examples (Lake et al., 2015; Wang et al., 2020).

Recent study (Doersch et al., 2020; Li et al., 2019) has disclosed that in the few-shot regime it is hard to learn abundant class-level knowledge for the novel classes, and instance-level information embodied by spatial parts of objects plays an important role. It has also been shown that dense, spatial features contain structural information, helpful for learning discriminative and transferable knowledge across categories (Zhang et al., 2020; Wertheimer et al., 2021; Xie et al., 2022; Liu et al., 2022). Among them, DeepEMD (Zhang et al., 2020; 2022) is of our particularity interest, as it is founded on a very effective statistical distance, i.e., Eather Mover's Distance (Rubner et al., 2000), also known as Wasserstein distance (Villani, 2008; Arjovsky et al., 2017) or Optimal Transport (Peyré & Cuturi, 2019). Fig. 1a depicts the idea of DeepEMD, where an image is represented by a set of local features indicated by points in the same color. The dis-similarity between the left (query) and right (support) images is formulated as matching of their corresponding features via EMD. The optimal matching flows indicated by solid lines are computed through QPTH solver (Amos & Kolter, 2017). Despite its competitive performance, DeepEMD is computationally very expensive due to inherent linear programming. More efficient EMD variants, e.g., Sinkhorn solver (Cuturi, 2013), can alleviate but fail to address this disadvantage, particularly for FSL problems at scale.

Towards addressing this problem, we propose in Sec. 4.2 a metric-based Gaussian EMD (GEMD-M) for FSL as depicted in Fig. 1b. Rather than discrete probability density function (PDF) in DeepEMD,

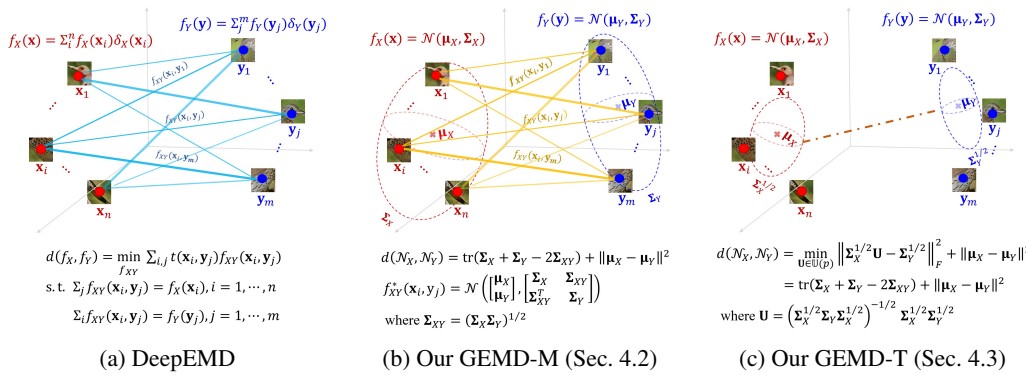

$$d(f_X, f_Y) = \min_{f_{XY}} \Sigma_{i,j} t(\mathbf{x}_i, \mathbf{y}_j) f_{XY}(\mathbf{x}_i, \mathbf{y}_j)$$
$$\text{s. t. } \Sigma_j f_{XY}(\mathbf{x}_i, \mathbf{y}_j) = f_X(\mathbf{x}_i), i = 1, \cdots, n$$
$$\Sigma_i f_{XY}(\mathbf{x}_i, \mathbf{y}_j) = f_Y(\mathbf{y}_j), j = 1, \cdots, m$$

(a) DeepEMD

$$d(\mathcal{N}_X, \mathcal{N}_Y) = \mathrm{tr}(\Sigma_X + \Sigma_Y - 2\Sigma_{XY}) + \|\mathbf{\mu}_X - \mathbf{\mu}_Y\|^2$$
$$f_{XY}^*(\mathbf{x}_i, \mathbf{y}_j) = \mathcal{N}\left(\begin{bmatrix} \mathbf{\mu}_X \\ \mathbf{\mu}_Y \end{bmatrix}, \begin{bmatrix} \Sigma_X & \Sigma_{XY} \\ \Sigma_{XY}^T & \Sigma_Y \end{bmatrix}\right)$$
$$\text{where } \Sigma_{XY} = (\Sigma_X \Sigma_Y)^{1/2}$$

(b) Our GEMD-M (Sec. 4.2)

$$d(\mathcal{N}_X, \mathcal{N}_Y) = \min_{\mathbf{U} \in \mathbb{U}(p)} \left\| \Sigma_X^{1/2} \mathbf{U} - \Sigma_Y^{1/2} \right\|_F^2 + \|\mathbf{\mu}_X - \mathbf{\mu}_Y\|^2$$
$$= \mathrm{tr}(\Sigma_X + \Sigma_Y - 2\Sigma_{XY}) + \|\mathbf{\mu}_X - \mathbf{\mu}_Y\|^2$$
$$\text{where } \mathbf{U} = \left(\Sigma_X^{1/2} \Sigma_Y \Sigma_X^{1/2}\right)^{-1/2} \Sigma_X^{1/2} \Sigma_Y$$

(c) Our GEMD-T (Sec. 4.3)

Figure 1: Illustration of DeepEMD versus our Gaussian EMD (GEMD) in 1-shot setting. The query (resp. support) image is designated by local features $\{\mathbf{x}_i\}_{i=1}^n$ (resp. $\{\mathbf{y}_j\}_{j=1}^m$) indicated by red (resp. blue) points and meanwhile by the corresponding patches; matching flows $f_{XY}(\mathbf{x}_i, \mathbf{y}_j)$ are indicated by solid lines. DeepEMD (a) formulates their dis-similarity as feature matching via EMD between discrete PDFs $f_X(\mathbf{x})$ and $f_Y(\mathbf{y})$ that involves a costly linear programming. We use Gaussians $\mathcal{N}(\mathbf{\mu}_\star, \Sigma_\star)$, $\star \in \{X, Y\}$ for modeling distributions, where $\mathbf{\mu}_\star$ and $\Sigma_\star$ are mean vector and covariance matrix, respectively. We propose metric-based GEMD (GEMD-M) (b), which computes EMD in closed form between Gaussian of the query and prototypical Gaussian of a support class. Furthermore, we propose a more efficient transfer learning-based method called GEMD-T (c), which learns a parametric EMD metric based on an equivalent but different form of EMD. Refer to Sec. 3 and Sec. 4 for detailed explanation of symbols and formulas.

we model feature distributions of images by Gaussian functions (Gaussians for short). We show that the optimal matching flows follow a joint Gaussian and can be expressed analytically; as a result, the EMD metric can be computed in closed form. GEMD-M constructs a nearest neighbor classifier that measures EMD between the Gaussian of the query image and prototypical Gaussian of a support class. However, GEMD-M is not that GPU-friendly as the involved distance is entangled, less suitable for parallel computation on GPU. To circumvent this problem, we further propose a transfer learning-based Gaussian EMD (GEMD-T) in Sec. 4.3, which is founded on an equivalent but different form of EMD for Gaussians as illustrated in Fig. 1c. The key idea of GEMD-T is to learn a parametric EMD in light of the square-roots of covariance matrices (via learnable orthogonal matrices) and mean vectors. The learnable metric is disentangled such that it can be implemented by a fully-connected layer followed by a softmax classifier, very suitable for GPU.

We experiment on large-scale Meta-Dataset (Triantafillou et al., 2020), which contains both in-domain and out-of-domain evaluations and builds realistic tasks of variable ways and shots. Also we experiment on 3 commonly used small-scale benchmarks in the Appendix. Our contributions are threefold. (1) We propose a metric-based FSL method (i.e., GEMD-M) that can compute EMD in closed form. Notably, we illuminate the metric amounts to feature matching, in which the optimal matching flows follow a joint Gaussian and can be expressed analytically. This formulation provides an intuitive understanding of the EMD metric for Gaussians. (2) We further present a more efficient transfer learning-based method called GEMD-T. Its core is to learn a parametric EMD via orthogonal matrices for a more discriminative metric. As far as we know, this is the first attempt that enables the EMD metric for Gaussians to be learned parametrically in deep learning. (3) Thorough ablation and extensive comparisons show the superiority of our GEMD. Particularly, on Meta-Dataset GEMD-T improves the previous best accuracies by 2.3% and 1.3% in the settings of training on ImageNet only and on multiple datasets, respectively.

## 2 RELATED WORKS

**Gaussians as image descriptors** Gaussian is the maximum entropy distribution for given mean and covariance among the family of continuous distributions (Bishop, 2006, Ch. 2). As analyzed in (Jaynes, 2003), Gaussian is a preferable PDF and enjoys ubiquitous applications due to its nature of maximum uncertainty with the least prior and many nice properties. Researchers widely used Gaussians for describing feature distributions of images in both traditional shallow learning (Nakayama et al., 2010; Li et al., 2017; Matsukawa et al., 2016) and deep learning (Wang et al.,

2017; Li et al., 2020). The Gaussian descriptors capture second-order statistics (i.e., covariance matrices), which is richer and more effective than lower order statistics (Sanchez et al., 2013). As the space of Gaussians is a Riemannian manifold (Rao, 1945), the classification performance can be greatly improved by leveraging the geometry (Li et al., 2017). Wang et al. (2017) propose a Gaussian embedding network based on theory of Lie group. Bilinear pooling (Lin et al., 2018) or covariance pooling (Wang et al., 2021; Song et al., 2023) yields second-order moments as image descriptors, which can be viewed as zero-mean Gaussians and shows better performance than global average pooling. In the FSL regime, Li et al. (2020) adopt Gaussian descriptors whose dis-similarity is measured by Kullback-Leibler (KL) divergence, which however is not a true metric (Csiszar, 1975). In contrast, our Gaussian EMD is a geodesic distance that can effectively leverage the geometry of Gaussian manifold (Malago et al., 2018).

**FSL leveraging dense features** In recent years a number of FSL methods propose to effectively leverage local features densely extracted by deep neural networks (DNNs). Recognizing the value of local features, Li et al. (2019) propose an image-to-class similarity measure. Doersch et al. (2020) disclose learning class-level knowledge is hard due to scarce training examples and instance-level spatial information plays a critical role. Wertheimer et al. (2021) formulate FSL as a Ridge regression problem, where the reconstruction error between features of a query image and those of every support class is used as distance measure. Liu et al. (2022) propose bidirectional random walk to learn mutual affiliations of bipartite dense features for FSL. DeepBDC (Xie et al., 2022) constructs Brownian distance covariance matrix based on Euclidean distances among all local features. DeepEMD (Zhang et al., 2020; 2022) formalizes image dis-similarity as optimal matching of local features via Earth Mover's Distance between discrete distributions. Our Gaussian EMD inherits the strength of DeepEMD (i.e., optimal feature matching), which can compute closed form distance metric and circumvent costly linear programming of EMD.

**Differential EMD** EMD (Rubner et al., 2000) has many applications in computer vision and machine learning (Peyré & Cuturi, 2019; Arjovsky et al., 2017). It involves a linear programming (LP) whose derivative is not straightforward to compute. Early attempt on differential EMD dates back to Zhao et al. (2010) that computes the gradient of EMD for object tracking. In deep learning, the QPTH solver (Amos & Kolter, 2017) develops a primal-dual interior point method for computing gradients of LP. QPTH enables DeepEMD (Zhang et al., 2020) to backpropagate gradients in DNNs, which unfortunately is computationally very expensive. The Sinkhorn solver (Cuturi, 2013) and its variants, e.g., IPOT (Xie et al., 2020), introduce entropy regularizers into LP, so the optimization can be accomplished with Sinkhorn-Knopp's algorithm (Sinkhorn & Knopp, 1967) that is fit for parallelization on GPU. In probability and statistics, it has long been known that EMD between Gaussians admits a closed form solution (Dowson & Landau, 1982; Givens & Shortt, 1984). Heusel et al. (2017) introduce Fréchet inception distance where Gaussian EMD is used to assess the quality of images produced by generative adversarial network. We propose Gaussian EMD for few-shot learning. We show that this metric amounts to optimal matching of image features, which is not elucidated before in deep learning to our best knowledge and can help intuitively interpret this metric; besides, we propose for the first time to learn a parametric EMD for Gausians.

## 3 DeepEMD in Retrospect

The idea of DeepEMD (Zhang et al., 2020; 2022) between a pair of images is shown in Fig. 1a. Let $\{\mathbf{x}_i\}_{i=1}^n$ and $\{\mathbf{y}_j\}_{j=1}^m$ respectively be local features of the left (query) and right (support) images extracted densely by DNNs. Statistically, EMD measures the discrepancy between two discrete distributions (Peyré & Cuturi, 2019, Sec. 2). Let $\{\mathbf{x}_i\}_{i=1}^n$ be observations of a random vector $X$ and let $f_X(\mathbf{x}) = \sum_i f_X(\mathbf{x}_i)\delta_X(\mathbf{x}_i)$ be its discrete PDF, where $f_X(\mathbf{x}_i)$ is probability of $\mathbf{x}_i$ and $\delta_X(\mathbf{x}_i)$ is Kronecker function that is equal to 1 if $\mathbf{x} = \mathbf{x}_i$ and 0 otherwise. Similarly, we denote by $f_Y(\mathbf{y}) = \sum_j f_Y(\mathbf{y}_j)\delta_Y(\mathbf{y}_j)$ the discrete PDF of $Y$. The image dis-similarity $d(f_X, f_Y)$ is defined as the following optimal transport problem:

$$d(f_X, f_Y) = \min_{f_{XY}} \sum_{i,j} t(\mathbf{x}_i, \mathbf{y}_j) f_{XY}(\mathbf{x}_i, \mathbf{y}_j), \tag{1}$$

where $f_{XY}$ satisfies $\sum_j f_{XY}(\mathbf{x}_i, \mathbf{y}_j) = f_X(\mathbf{x}_i), i = 1, \ldots, n$ and $\sum_i f_{XY}(\mathbf{x}_i, \mathbf{y}_j) = f_Y(\mathbf{y}_j), j = 1, \cdots, m$. Here $t(\mathbf{x}_i, \mathbf{y}_j)$ is transport cost per unit mass, often defined in terms of the Euclidean distance or inner product, and the matching flow $f_{XY}(\mathbf{x}_i, \mathbf{y}_j) \geq 0$ denotes the transport mass from $\mathbf{x}_i$ to $\mathbf{y}_j$. A cross-reference mechanism is developed to compute the discrete PDF.

From statistical view, EMD seeks an optimal joint distribution $f_{XY}^*(\mathbf{x}, \mathbf{y})$ that minimizes the total transport cost between marginals $f_X(\mathbf{x})$ and $f_Y(\mathbf{y})$. It can also be seen as feature matching between two set of features and $f_{XY}^*(\mathbf{x}, \mathbf{y})$ indicates the optimal matching flows. DeepEMD relies on differentiable QPTH solver (Amos & Kolter, 2017) for forward- and backward-propagations in a DNN, which however is computationally very expensive, infeasible to FSL at scale. More efficient Sinkhorn solver (Cuturi, 2013) and its variant (Xie et al., 2020) alleviate but still cannot address this disadvantage, as experimentally shown in Tab. 3e.

## 4  GAUSSIAN EMD FOR FEW-SHOT LEARNING

Few-shot classification is typically formulated as a multitude of episodes (tasks). Let $\mathbf{z}$ be an image and $c$ be its label. An episode consists of a support set $\mathcal{D}^s = \{(\mathbf{z}_i, c_i)\}_{i=1}^{|\mathcal{D}^s|}$ and a query set $\mathcal{D}^q = \{(\mathbf{z}_j, c_j)\}_{j=1}^{|\mathcal{D}^q|}$, where $|\cdot|$ denotes the set size. The task is to predict the labels of the query images given the support set with few training images per class. Notably, the numbers of classes and shots may vary from episode to episode, e.g., on Meta-Dataset (Triantafillou et al., 2020).

We adopt a two-stage pipeline of pre-training and meta-testing as in (Li et al., 2022; Tian et al., 2020; Dhillon et al., 2020). During pre-training, we train in a non-episodic manner a network using a softmax classifier with a cross-entropy loss that classifies all categories on the training set; then following (Tian et al., 2020), we adopt a self-distillation technique to improve the pre-trained model. In meta-testing, for each episode we use the pre-trained network as backbone on top of which we construct a classifier; next, we train the model using the support images.

In what follows, we first describe how to represent images by Gaussians. Then we introduce two instantiations, i.e., metric-based Gaussian EMD (GEMD-M) and transfer learning-based one (GEMD-T). Finally we give implementation of our methods.

### 4.1  IMAGE REPRESENTATIONS BY GAUSSIANS

To compute the Gaussians, we need to estimate mean vectors and covariance matrices of high-dimensional features extracted by a DNN. As the sizes of covariance matrices increase quadratically with feature dimension $p'$, for reducing computations we add a $1 \times 1$ convolution (conv) right after the last conv layer of the DNN, which decreases the dimension from $p'$ to a smaller $p$.

For an input image $\mathbf{z}$, we extract feature maps of size $h \times w \times p$ using the DNN parameterized by $\boldsymbol{\theta}$, where $h$ and $w$ denote spatial height and width and $p$ denotes the number of channels. We reshape the feature maps and obtain a set of $n = hw$ local features $\{\mathbf{x}_i\}_{i=1}^n, \mathbf{x}_i \in \mathbb{R}^p$, each of which is viewed as an observation of a random vector $X$. Note that $X$ is a function of $\boldsymbol{\theta}$ and $\mathbf{z}$. We use a Gaussian for modeling the distribution of $X$ as the descriptor of image $\mathbf{z}$:

$$\mathcal{N}(\boldsymbol{\mu}_X, \boldsymbol{\Sigma}_X) = \frac{1}{\sqrt{\det(2\pi\boldsymbol{\Sigma}_X)}} \exp\left(-\frac{1}{2}(\mathbf{x} - \boldsymbol{\mu}_X)^T \boldsymbol{\Sigma}_X^{-1}(\mathbf{x} - \boldsymbol{\mu}_X)\right), \tag{2}$$

where $\det$ denotes matrix determinant, $\boldsymbol{\mu}_X = \frac{1}{n}\sum_{i=1}^n \mathbf{x}_i$ and $\boldsymbol{\Sigma}_X = \frac{1}{n}\sum_{i=1}^n (\mathbf{x}_i - \boldsymbol{\mu}_X)(\mathbf{x}_i - \boldsymbol{\mu}_X)^T$ denote mean vector and covariance matrix that is symmetric positive definite (SPD), respectively. We mention that the space of Gaussians is a Riemannian manifold (Rao, 1945). For brevity, we shorten $\mathcal{N}(\boldsymbol{\mu}_X, \boldsymbol{\Sigma}_X)$ as $\mathcal{N}_X$ where necessary.

### 4.2  METRIC-BASED GAUSSIAN EMD

The idea of metric-based EMD (GEMD-M) in 1-shot setting is illustrated in Fig. 1b. Given a pair of left (query) and right (support) images, let $\{\mathbf{x}_i\}_{i=1}^n$ and $\{\mathbf{y}_j\}_{j=1}^m$ be their local features, respectively. As described previously, we estimate Gaussians $\mathcal{N}_X$ and $\mathcal{N}_Y$ to model their distributions. The dissimilarity of the two images is measured by EMD between the two Gaussians:

$$d(\mathcal{N}_X, \mathcal{N}_Y) = \inf_{f_{XY}} \int_{\mathbb{R}^p \times \mathbb{R}^p} \|\mathbf{x} - \mathbf{y}\|^2 f_{XY}(\mathbf{x}, \mathbf{y}) d\mathbf{x} d\mathbf{y}, \tag{3}$$

where $\|\cdot\|$ denotes the Euclidean distance and the joint distribution (matching flows) $f_{XY}(\mathbf{x}, \mathbf{y})$ is constrained to have fixed marginals, i.e., $\int_{\mathbb{R}^p} f_{XY}(\mathbf{x}, \mathbf{y}) d\mathbf{y} = \mathcal{N}_X$ and $\int_{\mathbb{R}^p} f_{XY}(\mathbf{x}, \mathbf{y}) d\mathbf{x} = \mathcal{N}_Y$. The following proposition describes the properties of $d(\mathcal{N}_X, \mathcal{N}_Y)$:

**Proposition 1** *For two Gaussians $\mathcal{N}_X$ and $\mathcal{N}_Y$, their EMD has the following closed form:*

$$d(\mathcal{N}_X, \mathcal{N}_Y) = \mathrm{tr}(\boldsymbol{\Sigma}_X + \boldsymbol{\Sigma}_Y - 2\boldsymbol{\Sigma}_{XY}) + \|\boldsymbol{\mu}_X - \boldsymbol{\mu}_Y\|^2, \tag{4}$$

*where $\mathrm{tr}$ denotes matrix trace and $\boldsymbol{\Sigma}_{XY} = (\boldsymbol{\Sigma}_X \boldsymbol{\Sigma}_Y)^{\frac{1}{2}}$ is square root of $\boldsymbol{\Sigma}_X \boldsymbol{\Sigma}_Y$. Besides, $f^*_{XY}(\mathbf{x}, \mathbf{y})$ that minimizes Eq. (3) follows a joint Gaussian:*

$$f^*_{XY}(\mathbf{x}, \mathbf{y}) = \mathcal{N}\left( \begin{bmatrix} \boldsymbol{\mu}_X \\ \boldsymbol{\mu}_Y \end{bmatrix}, \begin{bmatrix} \boldsymbol{\Sigma}_X & \boldsymbol{\Sigma}_{XY} \\ \boldsymbol{\Sigma}^T_{XY} & \boldsymbol{\Sigma}_Y \end{bmatrix} \right). \tag{5}$$

Derivation of Eqs. (4) and (5) can be found in (Dowson & Landau, 1982; Givens & Shortt, 1984).

Following the idea of ProtoNet (Snell et al., 2017), we construct a prototype for each class and then perform nearest neighbor classification. Let $\mathcal{N}(\bar{\boldsymbol{\mu}}_k, \bar{\boldsymbol{\Sigma}}_k)$ (simplified as $\bar{\mathcal{N}}_k$) be the prototypical Gaussian of the $k$-th support class, $k = 1, \ldots, K$. Here $\bar{\boldsymbol{\mu}}_k$ and $\bar{\boldsymbol{\Sigma}}_k$ are averages of the mean vectors and covariance matrices of all support images belonging to class $k$, respectively. By computing the distances between a query image and all support classes, we define the loss function as

$$l(\boldsymbol{\theta}, \{\bar{\mathcal{N}}_k\}_{k=1}^K) = -\frac{1}{|\mathcal{B}|} \sum_{(\mathbf{z}_i, c_i) \in \mathcal{B}} \log \frac{\exp(-d(\mathcal{N}_{X_i}, \bar{\mathcal{N}}_{c_i}))}{\sum_{k=1}^K \exp(-d(\mathcal{N}_{X_i}, \bar{\mathcal{N}}_k))}, \tag{6}$$

where $\boldsymbol{\theta}$ is the network parameters, $\mathcal{B}$ denotes a mini-batch and $\mathcal{N}_{X_i}$ denotes Gaussian descriptor of image $\mathbf{z}_i$ whose label is $c_i$. Similar to DeepEMD, we develop a structural layer where $\bar{\boldsymbol{\mu}}_k$ and $\bar{\boldsymbol{\Sigma}}_k$ are viewed as parameters for further finetuning.

From Eq. (4), it can be seen that $d(\mathcal{N}_X, \mathcal{N}_Y)$ compares mean vectors and covariance matrices of two Gaussians, which however is a lack of intuitive meaning. Fortunately, according to our formulation, $d(\mathcal{N}_X, \mathcal{N}_Y)$ can be naturally interpreted as optimal matching of dense features, in which the optimal matching flows $f^*_{XY}$ follow a joint Gaussian and can be expressed analytically. Fig. 1b intuitively illustrates how and why this metric works for a pair of images. This brings an illuminating interpretation of this well-known metric (Heusel et al., 2017), which is to our best knowledge not elucidated previously in deep learning. As $d(\mathcal{N}_X, \mathcal{N}_Y)$ is a Riemannian metric (Malago et al., 2018), our GEMD can capture the geodesic distance on the underlying manifold. In contrast, KL-divergence used in ADM (Li et al., 2020) is not a metric due to violation of the symmetry and triangle inequality (Csiszar, 1975), failing to consider the geometry of Gaussian manifold.

### 4.3 Transfer Learning-based Gaussian EMD

Despite the analytical form, the metric in GEMD-M (4) is entangled due to $\boldsymbol{\Sigma}_{XY}$ such that computation of a batch of distances is not that GPU-friendly. To circumvent this problem, we further propose transfer learning-based Gaussian EMD (GEMD-T) for FSL, which is founded on an equivalent but different form of EMD between two Gaussians as described by the following proposition.

**Proposition 2** *The EMD between $\mathcal{N}_X$ and $\mathcal{N}_Y$ is the solution to the following constrained minimization problem:*

$$d(\mathcal{N}_X, \mathcal{N}_Y) = \min_{\mathbf{U} \in \mathbb{U}(p)} \left\| \boldsymbol{\Sigma}_X^{\frac{1}{2}} \mathbf{U} - \boldsymbol{\Sigma}_Y^{\frac{1}{2}} \right\|_F^2 + \|\boldsymbol{\mu}_X - \boldsymbol{\mu}_Y\|^2, \tag{7}$$

*where $\|\cdot\|_F$ is Frobenius norm and $\mathbb{U}(p)$ is the space of $p \times p$ orthogonal matrices. The minimum is obtained when $\mathbf{U}$ is the factorizer of the polar decomposition of $\boldsymbol{\Sigma}_X^{\frac{1}{2}} \boldsymbol{\Sigma}_Y^{\frac{1}{2}}$, i.e., $\boldsymbol{\Sigma}_X^{\frac{1}{2}} \boldsymbol{\Sigma}_Y^{\frac{1}{2}} = (\boldsymbol{\Sigma}_X^{\frac{1}{2}} \boldsymbol{\Sigma}_Y \boldsymbol{\Sigma}_X^{\frac{1}{2}})^{\frac{1}{2}} \mathbf{U}$.*

The proof of this proposition is given in One may refer to (Bhatia et al., 2019) for its proof. This proposition provides an alternative view of the EMD between Gaussians. As shown in Fig. 1c, for a pair of images with Gaussian descriptors, this metric aims to align the square-root of the covariance matrix (via an orthogonal matrix) and mean vector of the left image with those of the right one.

Inspired by Prop. 2, we propose a transfer learning-based GEMD (GEMD-T). The key idea lies in a parametric EMD with learnable orthogonal matrices, aiming to learn a more discriminating EMD metric under supervision of the ground truth. Specifically, for an input image represented

by $\mathcal{N}(\boldsymbol{\mu}_X, \boldsymbol{\Sigma}_X)$, we learn an EMD metric to align $\boldsymbol{\Sigma}_X^{\frac{1}{2}}$ and $\boldsymbol{\mu}_X$ with a weight matrix $\mathbf{W}$ via an orthogonal matrix $\mathbf{U}$ and a weight vector $\mathbf{v}$ for every class. Here $\mathbf{W}$ and $\mathbf{v}$ can be regarded as the square-root of the covariance matrix and mean vector of the corresponding class, respectively. Then, we define the loss function of GEMD-T as:

$$l\big(\boldsymbol{\theta}, \{\mathbf{U}_k, \mathbf{W}_k, \mathbf{v}_k\}_{k=1}^K\big) = -\frac{1}{|\mathcal{B}|} \sum_{(\mathbf{z}_i, c_i) \in \mathcal{B}} \log \frac{\exp(\mathrm{tr}((\boldsymbol{\Sigma}_{X_i}^{\frac{1}{2}} \mathbf{U}_{c_i}) \mathbf{W}_{c_i}) + \boldsymbol{\mu}_{X_i}^T \mathbf{v}_{c_i})}{\sum_{k=1}^K \exp(\mathrm{tr}((\boldsymbol{\Sigma}_{X_i}^{\frac{1}{2}} \mathbf{U}_k) \mathbf{W}_k) + \boldsymbol{\mu}_{X_i}^T \mathbf{v}_k)}, \quad (8)$$

where $\mathbf{U}_k \in \mathbb{U}(p)$ is the orthogonal matrix to be learned, $\boldsymbol{\Sigma}_{X_i}$ and $\boldsymbol{\mu}_{X_i}$ denote the covariance matrix and mean vector of the Gaussian descriptor for image $\mathbf{z}_i$ whose label is $c_i$. Here we learn class-wise orthogonal matrices that are distinct for different classes. Alternatively, we can learn a class-share orthogonal matrix, i.e., $\mathbf{U}_1 = \cdots = \mathbf{U}_K$. The class-share scheme reduces significantly the number of orthogonal matrices to be learned. We compare the two schemes in Sec. 5.4. The matrix trace in Eq. (8) can be replaced by inner product of vectors. Let us denote $\mathbf{f}_i = [\mathrm{vec}(\mathbf{W}_1), \ldots, \mathrm{vec}(\mathbf{W}_K)]^T \mathrm{vec}(\boldsymbol{\Sigma}_{X_i}^{\frac{1}{2}} \mathbf{U}_{c_i}) + [\mathbf{v}_1, \ldots, \mathbf{v}_K]^T \boldsymbol{\mu}_{X_i}$, where $\mathrm{vec}(\cdot)$ indicates matrix vectorization. Then we can obtain a softmax-like classifier.

Clearly, the metric in GEMD-T is disentangled such that the classifier can be implemented by off-the-shelf fully-connected (FC) layer. This makes GEMD-T very suitable for parallel computation on GPU. It is worth mentioning that, as far as we know, we are among the first who learn the parametric EMD for better discriminating Gaussian distributions in deep learning.

## 4.4 IMPLEMENTATION OF GEMD

Our GEMD needs to compute square-roots of SPD matrices. A simple method for this is via eigen-decomposition (EIG) or singular value decomposition (SVD) (Lin & Maji, 2017), both of which unfortunately are GPU-unfriendly. So we adopt Newton-Schulz (NS) algorithm (Song et al., 2022; Li et al., 2018) that involves only matrix multiplications and additions and thus is every suitable for GPU. Given a SPD matrix $\boldsymbol{\Sigma}$, the NS algorithm consists of a pair of coupling iterations: $\mathbf{A} \leftarrow \frac{1}{2} \mathbf{A}(3\mathbf{I} - \mathbf{BA})$ and $\mathbf{B} \leftarrow \frac{1}{2}(3\mathbf{I} - \mathbf{BA})\mathbf{B}$, where $\mathbf{A}$ and $\mathbf{B}$ are initialized with $\boldsymbol{\Sigma}$ and the identity matrix $\mathbf{I}$, respectively. After some iterations $\mathbf{A}$ is approximately equal to $\boldsymbol{\Sigma}^{\frac{1}{2}}$. To ensure convergence of the NS algorithm, we divide $\boldsymbol{\Sigma}$ by $\mathrm{tr}(\boldsymbol{\Sigma})$ so that its matrix norm is less than 1; after iteration stop we multiply $\mathbf{A}$ by $\sqrt{\mathrm{tr}(\boldsymbol{\Sigma})}$ to recover the square-root of $\boldsymbol{\Sigma}$. Sec. 5.4 studies how iteration number affects GEMD.

For learning parametric EMD metric in GEMD-T, we are required to learn orthogonal matrices. The commonly used methods depend on some retraction operators, which map the update in the tangent space (Euclidean space) to the manifold of orthogonal matrices (Absil et al., 2007, Chap. 4). However, these retraction-based methods are not friendly to GPU due to complex matrix operations, e.g., matrix exponential or inversion. Therefore, we use the landing algorithm without retraction that has proven to be faster and more robust to numerical errors (Ablin & Peyré, 2022). Specifically, in order to learn an orthogonal matrix $\mathbf{U}$, during training process we make updates according to the following formula: $\mathbf{U} \leftarrow \mathbf{U} - \eta(\frac{1}{2}(\frac{\partial l}{\partial \mathbf{U}}\mathbf{U}^T - \mathbf{U}(\frac{\partial l}{\partial \mathbf{U}})^T) + \lambda(\mathbf{UU}^T - \mathbf{I}))\mathbf{U}$, where $\eta$ is a step size, $\lambda$ is a fixed parameter (set to 1), and $\frac{\partial l}{\partial \mathbf{U}}$ is the derivative of loss function $l$ with respect to $\mathbf{U}$.

## 5 EXPERIMENTS

In this section, we first introduce briefly the experimental setup. Then we compare with state-of-the-art (SOTA) methods on large-scale Meta-Dataset (Triantafillou et al., 2020). Finally, we conduct an ablation study for GEMD. In the Appendix, we describe detailed experimental settings, time complexity, visualization, and implementation of our counterparts as well as the additional experiments. The code and models of GEMD will be released.

## 5.1 EXPERIMENTAL SETUP

**Datasets** Meta-Dataset (Triantafillou et al., 2020) is a large-scale benchmark that involves both in-domain (In-D) and out-of-domain (Out-D) tasks. It initially consists of 10 datasets each designating

| | Test Domain | Matching-Net | BOHB | ProtoNet | Simple CANPS | ALFA+fo-Proto-MAML | FLUTE | TSA | RFS† | ADM† | DeepEMD† | GEMD-M (ours) | GEMD-T (ours) |
|---|---|---|---|---|---|---|---|---|---|---|---|---|---|
| In-D | ImageNet | 45.0±1.1 | 51.9±1.1 | 50.5±1.1 | 54.8±1.2 | 52.8±1.1 | 46.9±1.1 | 59.5±1.1 | 57.8±1.1 | 59.2±1.1 | 61.0±1.1 | 62.0±1.0 | **64.2±1.1** |
| Out-of-domain | Omniglot | 52.3±1.3 | 67.6±1.2 | 60.0±1.4 | 62.0±1.3 | 61.9±1.5 | 61.6±1.4 | 78.2±1.2 | 75.5±1.2 | 78.0±1.1 | 82.1±1.1 | 83.8±0.9 | **84.8±0.9** |
| | Aircraft | 49.0±0.9 | 54.1±0.9 | 53.1±1.0 | 49.2±0.9 | 63.4±1.1 | 48.5±1.0 | 72.2±1.0 | 68.3±1.1 | 69.3±1.0 | 67.0±1.0 | 72.8±1.1 | **81.7±1.0** |
| | Birds | 62.2±1.0 | 70.7±0.9 | 68.8±1.0 | 66.5±1.0 | 69.8±1.1 | 47.9±1.0 | 74.9±0.9 | 77.1±0.9 | 79.3±0.8 | 79.0±0.8 | 82.2±0.9 | **84.4±0.9** |
| | Textures | 64.2±0.9 | 68.3±0.8 | 66.6±0.8 | 71.6±0.7 | 70.8±0.9 | 63.8±0.8 | 77.3±0.7 | 75.7±0.7 | 74.5±0.7 | 72.4±0.7 | 74.8±0.7 | **78.6±0.7** |
| | Quick Draw | 42.9±1.1 | 50.3±1.0 | 49.0±1.1 | 56.6±1.0 | 59.2±1.2 | 57.5±1.0 | 67.6±0.9 | 62.6±1.0 | 64.3±1.0 | 64.4±0.9 | 68.6±0.8 | **72.4±0.9** |
| | Fungi | 34.0±1.0 | 41.4±1.1 | 39.7±1.1 | 37.5±1.2 | 41.5±1.2 | 31.8±1.0 | 44.7±1.0 | 46.6±1.2 | 44.6±1.0 | 45.9±1.1 | 49.9±1.2 | **50.6±1.1** |
| | VGG Flower | 80.1±0.7 | 87.3±0.6 | 85.3±0.8 | 82.1±0.9 | 86.0±0.8 | 80.1±0.9 | 90.9±0.6 | 90.2±0.6 | 91.1±0.6 | 89.1±0.7 | 91.8±0.8 | **94.8±0.5** |
| | Traffic Signs | 47.8±1.1 | 51.8±1.0 | 47.1±1.1 | 63.1±1.1 | 60.8±1.3 | 46.5±1.1 | **82.5±0.8** | 65.1±1.2 | 63.2±1.1 | 62.2±1.1 | 65.9±1.1 | 76.7±1.0 |
| | MSCOCO | 35.0±1.0 | 48.0±1.0 | 41.0±1.1 | 45.8±1.0 | 48.1±1.1 | 41.4±1.0 | **59.0±1.0** | 49.2±1.1 | 49.1±1.1 | 52.9±1.0 | 53.1±1.1 | 52.3±1.2 |
| | MNIST | – | – | – | – | – | 80.8±0.8 | 93.9±0.6 | 90.5±0.6 | 90.5±0.6 | 92.4±0.5 | 92.1±0.5 | **95.2±0.4** |
| | CIFAR-10 | – | – | – | – | – | 65.4±0.8 | **82.1±0.7** | 72.0±0.9 | 71.8±0.8 | 72.2±0.8 | 74.2±0.7 | 77.6±0.8 |
| | CIFAR-100 | – | – | – | – | – | 52.7±1.1 | **70.7±0.9** | 64.0±1.1 | 62.3±1.1 | 62.4±1.0 | 64.8±1.0 | 69.0±1.0 |
| Average | In-D | 45.0 | 51.9 | 50.5 | 54.8 | 52.8 | 46.9 | 59.5 | 57.8 | 59.2 | 61.0 | 62.0 | **64.2** |
| | Out-D | 51.9 | 59.9 | 56.7 | 59.4 | 62.4 | 56.5 | 74.5 | 69.7 | 69.8 | 70.2 | 72.8 | **76.5** |
| | All | 51.3 | 59.1 | 56.1 | 58.9 | 61.4 | 55.8 | 73.3 | 68.8 | 69.0 | 69.5 | 72.0 | **75.6** |
| | Rank | 11.5 | 8.1 | 9.8 | 8.7 | 7.8 | 9.9 | 2.8 | 4.9 | 4.9 | 4.5 | 2.8 | **1.5** |

† Our implementation.

Table 1: Comparison on Meta-Dataset in **SDL** setting (i.e., training on ImageNet only).

a different domain with distinct classes and distributions, i.e., ImageNet (Deng et al., 2009), Omniglot (Lake et al., 2015), Aircraft (Maji et al., 2013), Birds (Wah et al., 2011), Textures (Cimpoi et al., 2014), Quick Draw (Jongejan et al., 2016), Fungi (Schroeder & Cui, 2018), VGG Flower (Nilsback & Zisserman, 2008), Traffic Signs (Houben et al., 2013) and MSCOCO (Lin et al., 2014). As suggested in (Requeima et al., 2019; Triantafillou et al., 2021; Li et al., 2022), we include 3 extra datasets for meta-testing only, i.e., MNIST (LeCun & Cortes, 2010), CIFAR-10 (Krizhevsk, 2009) and CIFAR-100 (Krizhevsk, 2009). Following standard protocol, we conduct experiments in two settings, i.e., training on ImageNet only which we call single-domain learning (**SDL**) and training on 8 datasets called multi-domain learning (**MDL**).

**Implementation** For fair comparisons with previous works, we use ResNet-18 as backbone network. Similar to (Doersch et al., 2020; Li et al., 2020; Xie et al., 2022), we eliminate the last down-sampling of the backbone to obtain more spatial features. We follow the methodology of RFS (Tian et al., 2020), i.e., pre-training with self-distillation and meta-testing. During pre-training, we train the networks using standard softmax classifier with cross-entropy loss, which either classifies all categories on the train split of ImageNet in SDL setting or classifies all categories concatenated together on the train splits of 8 datasets in MDL setting; then, we use a sequential self-distillation technique to improve the models. In meta-testing, in SDL setting we freeze the backbone networks that are used to extract features for training the classifiers only; in MDL setting we finetune the whole networks, in which we only unfreeze the second convolutional layer of each residual block to reduce the risk of overfitting.

## 5.2 COMPARISON WITH SOTA METHODS IN SDL SETTING

In the single-domain learning (SDL), we train the network models on ImageNet only, while meta-testing on the test splits of ImageNet and 12 other datasets. The SOTA methods to be compared include MatchingNet (Vinyals et al., 2016), BOHB (Saikia et al., 2020), ProtoNet (Snell et al., 2017), Simple CANPS (Bateni et al., 2020), ALFA+fo-Proto-MAML (Baik et al., 2020), FLUTE (Triantafillou et al., 2021) and TSA (Li et al., 2022). Note that our three counterparts, i.e., RFS (Tian et al., 2020), ADM (Li et al., 2020) and DeepEMD (Zhang et al., 2020), did not experiment on Meta-Dataset. So we implement them ourselves using the same setup; refer to Appendix for implementation details .

As shown in Tab. 1, RFS achieves an 'Average All' accuracy of 68.8%, surpassing all previous SOTA methods but TSA. This suggests that the pre-trained backbone (embedding) is important for knowledge transfer. ADM is slightly better than RFS, achieving 0.2% higher accuracy. DeepEMD outperforms RFS by 0.7%, which we attribute to usage of dense features; in contrast, RFS simply performs global average pooling of all features. Our GEMD-M and GEMD-T improve DeepEMD by 2.5% and 6.1%, respectively. Notably, GEMD-T is much better than GEMD-M; we conjecture the reason may be that the entangled metric in GEMD-M makes network optimization difficult that greatly limits its performance. Compared to previous top-performer (i.e., TSA), GEMD-M has the same average rank though its accuracy is lower than TSA, while GEMD-T outperforming TSA by

| Test Domain | SUR | Simple CNAPs | URT | FLUTE | Tri-M | URL | TSA* | 2LM +TSA | RFS† | ADM† | DeepEMD† | GEMD-M (ours) | GEMD-T (ours) |
|---|---|---|---|---|---|---|---|---|---|---|---|---|---|
| **In-domain** | | | | | | | | | | | | | |
| ImageNet | 56.2±1.0 | 58.4±1.1 | 56.8±1.1 | 58.6±1.1 | 51.8±1.1 | 58.8±1.1 | 57.4±1.1 | 58.4±1.6 | 59.5±1.2 | 59.5±1.2 | 59.6±1.1 | 61.1±1.1 | **64.3**±1.0 |
| Omniglot | 94.1±0.4 | 91.6±0.6 | 94.2±0.4 | 92.0±0.6 | 93.2±0.5 | 94.5±0.4 | 95.0±0.4 | **95.4**±0.8 | 93.9±0.5 | 94.7±0.4 | 95.0±0.4 | 95.2±0.4 | 95.3±0.7 |
| Aircraft | 85.5±0.5 | 82.0±0.7 | 85.8±0.5 | 82.8±0.7 | 87.2±0.5 | 89.4±0.4 | 89.3±0.4 | 89.3±0.5 | 91.3±0.6 | 92.2±0.4 | 91.8±0.4 | 91.4±0.5 | **93.4**±0.4 |
| Birds | 71.0±1.0 | 74.8±0.9 | 76.2±0.8 | 75.3±0.8 | 79.2±0.8 | 80.7±0.4 | 81.4±0.7 | 82.1±0.7 | 84.9±0.7 | 84.8±0.7 | 85.0±0.6 | 86.4±0.8 | **87.8**±0.7 |
| Textures | 71.0±0.8 | 68.8±0.9 | 71.6±0.7 | 71.2±0.8 | 68.8±0.8 | 77.2±0.7 | 76.7±0.7 | **78.2**±1.0 | 77.4±0.9 | 77.4±0.7 | 76.4±0.7 | 76.7±0.9 | 78.1±0.8 |
| Quick Draw | 81.8±0.6 | 76.5±0.8 | 82.4±0.6 | 77.3±0.7 | 79.5±0.7 | 82.5±0.6 | 76.7±0.7 | 82.8±0.6 | 79.6±0.7 | 82.1±0.6 | 81.4±0.6 | 82.3±0.6 | **83.1**±0.7 |
| Fungi | 64.3±0.9 | 46.6±1.0 | 64.0±1.0 | 48.5±1.0 | 58.1±1.1 | 68.1±0.9 | 67.4±1.0 | 69.5±1.2 | 62.0±1.1 | 66.6±1.0 | **70.5**±1.0 | 62.7±1.2 | 64.0±1.1 |
| VGG Flower | 82.9±0.8 | 90.5±0.5 | 87.9±0.6 | 90.5±0.5 | 91.6±0.6 | 92.0±0.5 | 92.2±0.5 | 92.4±1.6 | 93.3±0.7 | 92.5±0.6 | 91.6±0.5 | 92.3±0.7 | **93.8**±0.6 |
| **Out-of-domain** | | | | | | | | | | | | | |
| Traffic Signs | 51.0±1.1 | 57.2±1.0 | 48.2±1.1 | 63.0±1.0 | 58.4±1.1 | 63.3±1.1 | 83.5±0.9 | **88.4**±2.1 | 79.4±1.0 | 74.1±1.1 | 71.0±1.1 | 83.5±0.9 | 87.0±0.8 |
| MSCOCO | 52.0±1.1 | 48.9±1.1 | 51.5±1.1 | 52.8±1.1 | 50.0±1.1 | **57.3**±1.1 | 55.8±1.1 | **57.3**±1.5 | 50.7±1.0 | 50.5±1.2 | 54.8±1.1 | 54.5±1.2 | 53.7±1.1 |
| MNIST | 94.3±0.4 | 94.6±0.4 | 90.6±0.5 | 96.2±0.3 | 95.6±0.5 | 94.7±0.4 | 96.7±0.4 | 97.3±1.2 | 96.2±0.5 | 95.4±0.4 | 95.6±0.4 | 97.0±0.4 | **97.4**±0.5 |
| CIFAR-10 | 66.5±0.9 | 74.9±0.7 | 67.0±0.8 | 75.4±0.6 | 78.6±0.7 | 74.2±0.6 | **80.6**±0.4 | 76.5±1.4 | 77.0±0.9 | 78.4±0.9 | 78.6±0.8 | 78.9±0.9 | 79.6±0.9 |
| CIFAR-100 | 56.9±1.1 | 61.3±1.1 | 57.3±1.0 | 62.0±1.0 | 67.1±1.0 | 63.5±1.0 | 69.6±1.0 | 67.7±1.5 | 68.0±1.1 | 66.4±1.1 | 67.0±1.1 | 67.7±1.1 | **72.4**±1.0 |
| **Average** | | | | | | | | | | | | | |
| In-D | 75.9 | 73.7 | 77.4 | 74.5 | 76.2 | 80.4 | 80.2 | 80.9 | 80.2 | 81.2 | 81.4 | 81.0 | **82.5** |
| Out-D | 64.1 | 67.4 | 62.9 | 69.9 | 69.9 | 70.6 | 77.2 | 77.4 | 74.3 | 73.0 | 73.4 | 76.3 | **78.0** |
| All | 71.4 | 71.2 | 71.8 | 72.7 | 73.8 | 76.6 | 79.0 | 79.5 | 77.9 | 78.1 | 78.3 | 79.2 | **80.8** |
| Rank | 10.5 | 11.3 | 10.1 | 9.9 | 9.5 | 6.7 | 5.0 | 4.0 | 6.0 | 5.5 | 5.5 | 4.5 | **2.5** |

*Corrected results by TSA's authors (see the official supplement). †Our implementation.

Table 2: Comparison on Meta-Dataset in **MDL** setting (i.e., training on eight datasets).

2.3%. Our GEMD-T takes the lead on 9 out of 13 datasets, generally by large margins, achieving the first position in average rank across the board.

## 5.3 COMPARISON WITH SOTA METHODS IN MDL SETTING

In the multi-domain learning (MDL), the models are trained on 8 datasets with diverse distributions, and are meta-tested on the test splits of them together with those of the other 5 datasets. We contrast our methods with SUR (Dvornik et al., 2020), Simple CNAPs (Bateni et al., 2020), URT (Liu et al., 2021a), FLUTE (Triantafillou et al., 2021), tri-M (Liu et al., 2021b), URL (Li et al., 2021), TSA (Li et al., 2022), 2LM+TSA (Qin et al., 2023). As in the SDL setting, we also implement the three counterparts, i.e., RFS, ADM and DeepEMD. Note that all methods above are inductive and we do not compare with high-performance transductive methods, e.g., Tao et al. (2023), as they use the query images during training.

Tab. 2 shows the comparison results. We can see that RFS is still a strong baseline, which achieves 77.9% in 'Average All' accuracy, higher than most of the previous arts except TSA and 2LM+TSA. As for our counterparts, ADM and DeepEMD perform marginally better than RFS. Our GEMD-M improves DeepEMD by 0.9% and GEMD-T outperforms it by 2.5%. Note that previous top-competitor (i.e., TSA) uses pre-trained model of URL, where a universal model is distilled from 8 networks separately trained on individual datasets; 2LM+TSA improves TSA and achieves previously the best result. Compared to them, our EMD-M performs comparatively while our GEMD-T surpasses 2LM+TSA by 1.3%, taking the lead in average rank across the board.

## 5.4 ABLATION STUDY

We conduct an ablation study on key components of our GEMD. To facilitate fast evaluations involving a great number of experiments, we ablate in SDL setting (i.e., train on ImageNet only).

**Dimension reduction** As described in Sec. 4.1, we insert a 1×1 convolution for dimension reduction (DR) after the last conv layer of the backbone, decreasing feature dimension from $p' = 512$ to a smaller $p$. Tab. 3a presents the accuracy and average wall-clock time per episode of GEMD versus $p$, where the iteration number of NS algorithm is fixed to five and class-share scheme is used in GEMD-T. Since the ways and shots are changing from episode to episode, to facilitate time measurement, we select randomly a hundred of 20-way 10-shot episodes with 10 queries per class on validation set of ImageNet and report average time per task using GeForce RTX 3090. It can be seen that, starting from $p = 128$, the accuracy of GEMD-T increases and saturates around $p = 192$, and GEMD-M demonstrates a similar trend. We note that with same dimension GEMD-T performs significantly better than GEMD-M while running much faster. This suggests that our transfer learning-based method has better generalization capability than the metric-based one. To tradeoff between the performance and speed, we set throughout $p$ to 128 and 192 for GEMD-M and GEMD-T, respectively.

| Dim $(p)$ | GEMD-M Acc | GEMD-M Time | GEMD-T Acc | GEMD-T Time |
|---|---|---|---|---|
| 320 | 71.6 | 1434 | 75.4 | 88 |
| 256 | 71.7 | 715 | 75.7 | 70 |
| 192 | 71.5 | 488 | 75.6 | 64 |
| 160 | 70.8 | 388 | 74.4 | 62 |
| 128 | 70.4 | 197 | 74.2 | 58 |

(a) Dimension reduction

| NS iter $(r)$ | GEMD-M Acc | GEMD-M Time | GEMD-T Acc | GEMD-T Time |
|---|---|---|---|---|
| 3 | 68.2 | 149 | 74.9 | 62 |
| 5 | 70.4 | 197 | 75.6 | 64 |
| 7 | 71.5 | 236 | 75.5 | 68 |
| 10 | 72.0 | 328 | 75.3 | 78 |
| 15 | 71.9 | 414 | 75.0 | 88 |

(b) Newton-Schulz algorithm

| Scheme | GEMD-T |
|---|---|
| Class-wise | 75.2 |
| Class-share | 75.6 |

(c) Learning $\mathbf{U}_i$

| $h \times w$ | GEMD-M | GEMD-T |
|---|---|---|
| $6 \times 6$ | 70.4 | 74.2 |
| $11 \times 11$ | 72.0 | 75.6 |

(d) Size of feature maps

| | Method | Acc | Time |
|---|---|---|---|
| DeepEMD | QPTH | – | >700K |
| | Sinkhorn | 69.5 | 420 |
| | IPOT | 69.4 | 540 |
| Ours | GEMD-M | 72.0 | 328 |
| | GEMD-T | 75.6 | 64 |
| | RFS | 68.8 | 60 |
| | TSA | 73.3 | 120 |
| | ProtoNet | 56.1 | 45 |

(e) Speed of GEMD

Table 3: Ablation study on Meta-Dataset in SDL setting. We report accuracy (Acc, %) of 'Average All' and wall-clock time (ms) per episode with GeForce RTX 3090.

**Newton-Schulz (NS) algorithm** As Sec. 4.4 explains, our methods use NS algorithm for computing square-roots of SPD matrices. Tab. 3b shows how the number $r$ of NS iterations affects the performance of GEMD. As $r$ increases, the accuracy of GEMD-T reaches a peak value at $r = 5$ and then decreases slightly, and meanwhile the wall-clock time steadily yet mildly enlarges. The accuracy of GEMD-M improves continually until $r = 10$; however, the increment of the wall-clock time is considerable. We set the number of iterations $r$ to 5 for GEMD-T and 10 for GEMD-M throughout the paper, unless otherwise specified.

**Learning orthogonal matrix $\mathbf{U}_i$ in GEMD-T** Sec. 4.3 introduces two schemes to learn the parameters of orthogonal matrices using the landing algorithm (cf. Sec. 4.4), i.e., the class-share scheme and class-wise scheme. As shown in Tab. 3c, the class-share scheme performs better than the class-wise one, though the latter might seem to be more promising as it learns category-specific orthogonal matrices. We hypothesize that the large number of orthogonal matrices in the class-wise scheme makes optimization difficult and brings side effect. So we opt for the class-share scheme.

**Size of feature maps** To obtain more local features, we remove down-sampling (DS) of the last stage in ResNet. As such, the size (i.e., $h \times w$) of feature maps increases from original $6 \times 6$ with DS to $11 \times 11$ without DS. As seen in Fig. 3d, removal of DS brings an improvement of 1.4%~1.6%. This suggests that more local features are helpful for learning more discriminative and transferable knowledge. We thus prefer without DS across the paper.

**Wall-clock time comparison** The original DeepEMD relying on differentiable QPTH solver (Amos & Kolter, 2017) is computationally prohibitive on Meta-Dataset. So we turn to more efficient Sinkhorn solver (Cuturi, 2013) and one of its variants called IPOT solver (Xie et al., 2020). As shown in Tab. 3e, QPTH is extremely expensive while both Sinkhorn and IPOT are computationally more efficient. DeepEMD with Sinkhorn and IPOT solvers achieve similar performance; our GEMD-M outperforms them by ∼1.5% with comparable speed. Notably, our GEMD-T performs significantly better, improving DeepEMD by ∼6% while running 6 times faster. We also compare to RFS, TSA and ProtoNet. Our GEMD-T is clearly superior to RFS while having almost the same speed. Compared to state-of-the-art TSA, our GEMD-T is 2 times faster and meanwhile performs better. ProtoNet consists of pre-training plus meta-training along with meta-test where no training is required. Therefore, ProtoNet runs the fastest, but with significantly inferior performance.

# 6 CONCLUSION

Our GEMD inherits the strength of DeepEMD, i.e., optimal matching of dense features that helps representation learning in the few-shot regime. As GEMD uses Gaussian descriptors, it can capture second-order statistics (i.e., covariance matrices), which is richer and has better representational ability than lower order discrete statistics (Sanchez et al., 2013) used in DeepEMD. Moreover, our metric is a geodesic distance admitting closed form expression, so it can effectively exploit the geometry of Gaussian manifold. As a result, our GEMD is much faster and stronger than DeepEMD, while being very compelling compared to state-of-the-art methods. The proposed GEMD is applicable to other vision tasks such as few-shot action recognition and image retrieval.

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

# A APPENDIX

## A.1 STATISTICS OF META-DATASET AND EVALUATION PROTOCOLS

Meta-Dataset (Triantafillou et al., 2020) initially consists of 10 datasets. As in (Requeima et al., 2019; Triantafillou et al., 2021; Li et al., 2022), we introduce 3 additional datasets for meta-testing only. It is a large-scale benchmark, span diverse data sources, and contains realistic tasks. Tab. 4a summarizes statistics of 13 datasets each of which designates a different domain. Notably, both the number of ways and shots in Meta-Dataset are variable from episode to episode that leads to class imbalance. These tasks are more realistic than those of previous FSL benchmarks, e.g., *mini*ImageNet (Vinyals et al., 2016) and *tired*ImageNet (Ren et al., 2018), where typically 5-way 1-shot/5-shot tasks are constructed. The input images are $84 \times 84$ for both training and test. Tab. 4b shows two protocols specified on Meta-Dataset, i.e., train on ImageNet only that we call single-domain learning (SDL) and train on multiple datasets called multi-domain learning (MDL). Both SDL and MDL contain in-domain (In-D) and out-of-domain (Out-D) evaluations. For meta-testing, we randomly sample 600 tasks per dataset and report average accuracy and 95% confidence score. We also report average accuracies of In-D/Out-D/all domains and average rank across the board.

## A.2 IMPLEMENTATION DETAILS OF GEMD ON META-DATASET

Following (Li et al., 2022; Tian et al., 2020; Dhillon et al., 2020), we adopt a two-stage pipeline for few-shot learning, i.e., pre-training in a non-episodic manner and meta-testing. In both stages, we use the SGD optimizer with momentum (set to 0.9 throughout). For fair comparisons with previous methods, we use ResNet-18 architecture (He et al., 2016) slightly modified as in (Requeima et al., 2019; Bateni et al., 2020). Regarding Netwon-Schulz (NS) algorithm that computes square-roots of symmetric positive definite (SPD) matrices, we use the code released by Li et al. (2018). For the landing algorithm that learns the orthogonal matrices in GEMD-T, we adopt the implementation by Ablin & Peyré (2022) that is public available.

### A.2.1 PRE-TRAINING STAGE

In this stage, we train networks non-episodically with standard softmax classifiers and cross-entropy losses. Following the methodology of RFS (Tian et al., 2020), we adopt a self-distillation technique of multiple generations (one generation in our case) to boost the pre-trained models, whose implementation is based on the code released by the authors of RFS. We use the same augmentation as (Dvornik et al., 2020; Li et al., 2021), i.e., random crop and color jittering. We add a dropout ($\text{rate} = 0.5$) in the last fully-connected (FC) layer. The weight decay of SGD is set to 0.0001.

**SDL** The networks are trained on 712 categories of the train split of ImageNet, and the best model is selected on the validation split. We train the networks up to 600K iterations with a batch size of 64. The initial learning rate is 0.05, divided by 10 after 350K and 500K iterations, respectively.

**MDL** We train the models on the train splits of 8 datasets and select the best model using their validation splits. We concatenate the categories of all 8 datasets and train a standard multi-way classifier. Following Li et al. (2021), we randomly select $64 \times 7$ images on ImageNet and 64 for each of the other 7 datasets in forming a mini-batch. The learning rate starts from 0.35 and decays according to a cosine annealing scheduler (He et al., 2019) within 248K iterations.

### A.2.2 META-TESTING STAGE

For each episode, we build a classifier with the pre-trained network as the backbone, training with a constant learning rate on the support set and predicting labels of the query set. The mini-batch of GEMD-T is composed of all support images.

**SDL** For each episode, we freeze the backbone of the pre-trained model and train the classifier only. In GEMD-M, the nearest neighbor classifier is initialized with the protypical Gaussians of support classes and then trained in 5 epochs with a learning rate of 0.001, a weight decay of 0.0001 and a batch size of 64. For GEMD-T, the softmax-like classifier is trained from scratch for 600 iterations with a learning rate of 0.01 and a weight decay of 0.0005.

**(a) Statistics**

| Dataset (domain) | Image Num | Train | Val | Test |
|---|---|---|---|---|
| ImageNet (Deng et al.) | 1.3M | 712 | 158 | 130 |
| Omniglot (Lake et al.) | 32K | 883 | 81 | 659 |
| Aircraft (Maji et al.) | 10K | 70 | 15 | 15 |
| Birds (Wah et al.) | 12K | 140 | 30 | 30 |
| Texture (Cimpoi et al.) | 6K | 33 | 7 | 7 |
| Quick Draw (Jongejan et al.) | 50M | 241 | 52 | 52 |
| Fungi (Schroeder & Cui) | 100K | 994 | 200 | 200 |
| VGG Flower (Nilsback & Zisserman) | 7K | 71 | 15 | 16 |
| MSCOCO (Lin et al.) | 860K | | 40 | 40 |
| Traffic Signs (Houben et al.) | 39K | | | 43 |
| MNIST (LeCun & Cortes) | 70K | | | 10 |
| CIFAR-10 (Krizhevsk) | 60K | | | 10 |
| CIFAR-100 (Krizhevsk) | 60K | | | 100 |
| Input size | 84×84 | | | |
| Tasks | **variable-way & -shot** | | | |

**(b) Evaluation protocols**

| SDL (train on ImageNet only) | | | MDL (train on eight datasets) | | |
|---|---|---|---|---|---|
| Train | Val | Test | Train | Val | Test |
| ImageNet | ImageNet | ImageNet | ImageNet | ImageNet | ImageNet |
| | | Omniglot | Omniglot | Omniglot | Omniglot |
| | | Aircraft | Aircraft | Aircraft | Aircraft |
| | | Birds | Birds | Birds | Birds |
| | | Texture | Texture | Texture | Texture |
| | | Quick Draw | Quick Draw | Quick Draw | Quick Draw |
| | | Fungi | Fungi | Fungi | Fungi |
| | | VGG Flower | VGG Flower | VGG Flower | VGG Flower |
| | | MSCOCO | | | MSCOCO |
| | | Traffic Signs | | | Traffic Signs |
| | | MNIST | | | MNIST |
| | | CIFAR-10 | | | CIFAR-10 |
| | | CIFAR-100 | | | CIFAR-100 |

Table 4: Statistics of individual datasets on Meta-Dataset (a) and evaluation protocols of single-domain learning (SDL) and multi-domain learning (MDL) (b).

**MDL** We finetune the model with a weight decay of 0.0001, in which we only unfreeze the second conv layer of every residual block to avoid the risk of overfitting. For GEMD-M, after initializing the classifier with the prototypes of support classes, we finetune the network for 50 epochs with a learning rate of 0.0001 and a batch size of 32. For GEMD-T, we adopt a dropout ($\text{rate}=0.7$) for the last FC layer; the network is finetuned for 70 epochs with the learning rate set to 0.005.

### A.2.3 ANALYSIS OF TIME COMPLEXITY

The dominant operations of GEMD are matrix square-roots computed via NS iterations. Let $n$ be the number of features and $p$ be feature dimension. GEMD-M is a metric-based method that has a complexity of $O(KQr_{gm}p^3)$ for an episode with $K$-ways and $Q$-queries, where $r_{gm}$ is the number of NS iterations. GEMD-T is a transfer learning-based method whose complexity is $O(|\mathcal{B}|r_{gt}p^3)$ where $|\mathcal{B}|$ is batch size and $r_{gt}$ is iteration number. The DeepEMD with QPTH solver runs in a time of $O(KQn^3 \log n)$ (Zhang et al., 2022; Amos & Kolter, 2017). Both the Sinkhorn and IPOT solvers are based on Sinkhorn-Knopp algorithm (Sinkhorn & Knopp, 1967) and have a provable complexity of $O(KQn^2 \log n \|\mathbf{T}\|_\infty^3 \varepsilon^{-3})$ (Altschuler et al., 2017), where $\|\mathbf{T}\|_\infty$ is the infinity norm of matrix $\mathbf{T} = [t(\mathbf{x}_i, \mathbf{y}_j)]$ (cf. Sec. 3), and $\varepsilon > 0$ is the difference between the value of exact EMD and that of the approximate one. Both of them generally require iterations far larger than GEMD.

The QPTH solver has a high time complexity. Besides, it is unfit for GPU parallel computing, as it involves inverse of normal matrix due to the used interior point algorithm (Luenberger & Ye, 2016). In practice, GEMD-M and DeepEMD with Sinkhorn (or IPOT) solver have comparable complexities and are much more efficient than QPTH solver. Compared to them, though having similar complexity, GEMD-T is significantly faster as it can be implemented via off-the-shelf FC layer, highly suitable for GPU. The wall-clock time in Sec. 5.4 validates the analysis above.

### A.3 VISUALIZATION OF GEMD

We visualize Gaussian EMD (GEMD) from two aspects. First, as our GEMD-M is a metric based method that essentially performs matching of features (patches), we visualize the optimal patches matched between the query image and support image. Second, our GEMD-T is transfer learning based method and therefore we adopt commonly used heatmaps to see the effect of its representation learning.

**Visualization by Patch Matching** We visualize the effect of patch matching and compare with DeepEMD in the 1-shot setting. To this end, given a support image, for every query image we compute the optimal matching flows between them using GEMD-M and DeepEMD, respectively. For clarity, we only show matching results of some typical patches that are indicated by different color squares. Specifically, for a color patch in the support image, its best match that has the largest matching flow is determined in the query image and is indicated by a square with the same color.

Fig. 2 shows matching results of four categories, i.e., mushroom, arctic fox, school bus and beer bottle. It can be seen that, overall, the matches determined by GEMD-M are more accurate than those of DeepEMD, which we attribute to more discriminative capability of GEMD. It is worth noting that, when performing few-shot recognition, we directly compute closed form EMD between Gaussians without explicitly computing the optimal matching flows; in contrast, only after computing the optimal matching flows via expensive linear programming, can one obtain the distance in DeepEMD.

**Visualization by Heatmaps** We visualize in Fig. 3 the heapmaps of images by CAM (Zhou et al., 2016). We select four categories, i.e., Tench, Daisy, Rosehip and Meekat, for visualization with models learned by GEMD-T and DeepEMD. Let us take Tench as an example. From Fig. 3a, it can be seen that GEMD can accurately focus on the regions of the tench while neglecting the backgrounds. In contrast, DeepEMD has larger tendency to be distracted by backgrounds; specifically, it seems that the model largely attends to both the objects and the person who holds it. We have similar observations for the images of the other categories. The comparison above indicates that GEMD can learn more discriminative features than DeepEMD, demonstrating better capability of representation learning. We attribute this superiority over DeepEMD to the Gaussian descriptor that can capture higher order statistics and to the metric that is a geodesic distance and can effectively leverage geometry of the underlying manifold.

## A.4 IMPLEMENTATION OF OUR COUNTERPARTS

This section first introduces how to implement DeepEMD with Sinkhorn solver (Cuturi, 2013) and with IPOT solver (Xie et al., 2020). Subsequently, we provide implementation of RFS (Tian et al., 2020) and ADM (Li et al., 2020). For a fair comparison, in MDL setting we finetune the backbones of the three counterparts with the same finetuning strategy.

### A.4.1 IMPLEMENTATION OF DEEPEMD

During training, DeepEMD (Zhang et al., 2020) adopt the differentiable QPTH solver (Amos & Kolter, 2017) for the linear programming (LP) problem. The QPTH solver enables forward- and backward-propagations of EMD in DNN, but is so costly as to be infeasible for large-scale Meta-Dataset. For acceleration, Zhang et al. (2020) propose to use the OpenCV solver (Kaehler & Bradski, 2017) that however cannot compute gradients for backpropagation. So we resort to efficient Sinkhorn and IPOT solvers.

Let us keep the notations of the main paper in mind. The Sinkhorn solver aims to optimize the following entropy regularized EMD:

$$\min_{f_{XY}} \sum_{i,j} t(\mathbf{x}_i, \mathbf{y}_j) f_{XY}(\mathbf{x}_i, \mathbf{y}_j) + \alpha \sum_{i,j} f_{XY}(\mathbf{x}_i, \mathbf{y}_j) \log f_{XY}(\mathbf{x}_i, \mathbf{y}_j), \tag{9}$$

where $\alpha$ is a regularizing parameter. The discrete joint distribution $f_{XY}(\mathbf{x}, \mathbf{y})$ is constrained to have fixed marginals, i.e., $\sum_j f_{XY}(\mathbf{x}_i, \mathbf{y}_j) = f_X(\mathbf{x}_i)$ and $\sum_i f_{XY}(\mathbf{x}_i, \mathbf{y}_j) = f_Y(\mathbf{y}_j)$ for feasible values of $i$ and $j$. Thanks to the entropic regularizer, i.e., the second term in (9), the objective can be minimized with the Sinkhorn-Knopp's matrix scaling algorithm (Sinkhorn & Knopp, 1967) that is appropriate to parallelization on GPU. As suggested in (Xie et al., 2020), $\alpha$ cannot be very small; otherwise, it may lead to numerical instability.

For IPOT method, in the $k$-th iteration, the original LP problem is constrained by entropy-based Bregman divergence and the objective can be written as:

$$\min_{f_{XY}} \sum_{i,j} (t(\mathbf{x}_i, \mathbf{y}_j) - \beta \log f_{XY}^{(k)}(\mathbf{x}_i, \mathbf{y}_j)) f_{XY}(\mathbf{x}_i, \mathbf{y}_j) + \beta \sum_{i,j} f_{XY}(\mathbf{x}_i, \mathbf{y}_j) \log f_{XY}(\mathbf{x}_i, \mathbf{y}_j) \tag{10}$$

where $f_{XY}^{(k)}(\mathbf{x}_i, \mathbf{y}_j)$ denotes the joint distribution at iteration $k$, and $f_{XY}(\mathbf{x}, \mathbf{y})$ is constrained to have fixed marginals $f_X(\mathbf{x})$ and $f_Y(\mathbf{y})$. Naturally, the objective (10) can be solved by Sinkhorn-Knopp's algorithm, and, as suggested in (Xie et al., 2020), one iteration is sufficient for this inner loop. Consequently, one can obtain the joint distribution at $(k+1)$-th iteration, i.e., $f_{XY}^{(k+1)}(\mathbf{x}_i, \mathbf{y}_j)$. The process is repeated until convergence. Compared to the Sinkhorn solver, the regularizing parameter $\beta$ can be selected in a larger range.

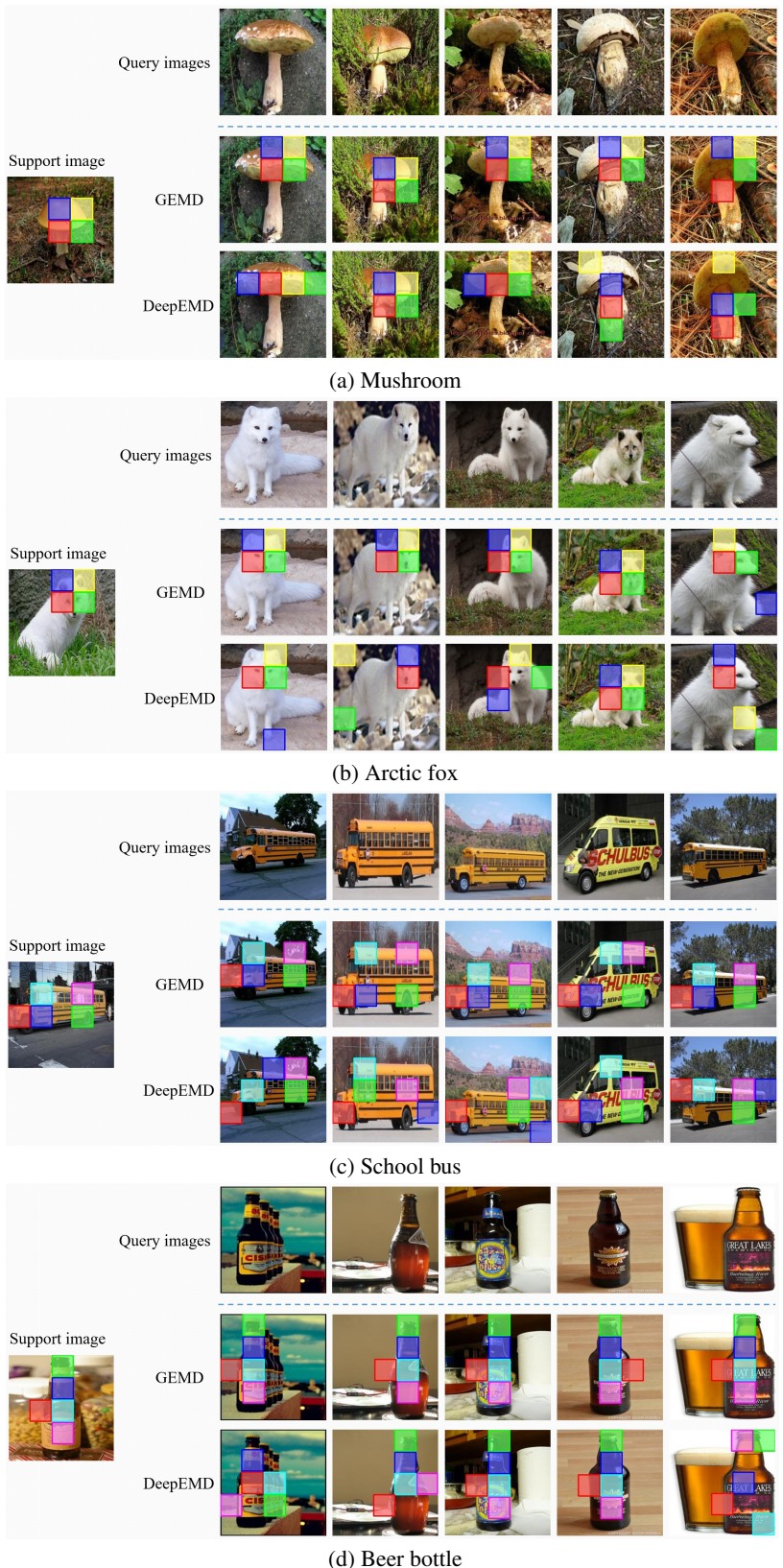

(a) Mushroom

(b) Arctic fox

(c) School bus

(d) Beer bottle

Figure 2: Visualization of matching of typical patches with GEMD-M in 1-shot setting. For one patch indicated by a colored square in a support image (left), its best match in each query image (right) is shown by an identically colored square. Best viewed in color.

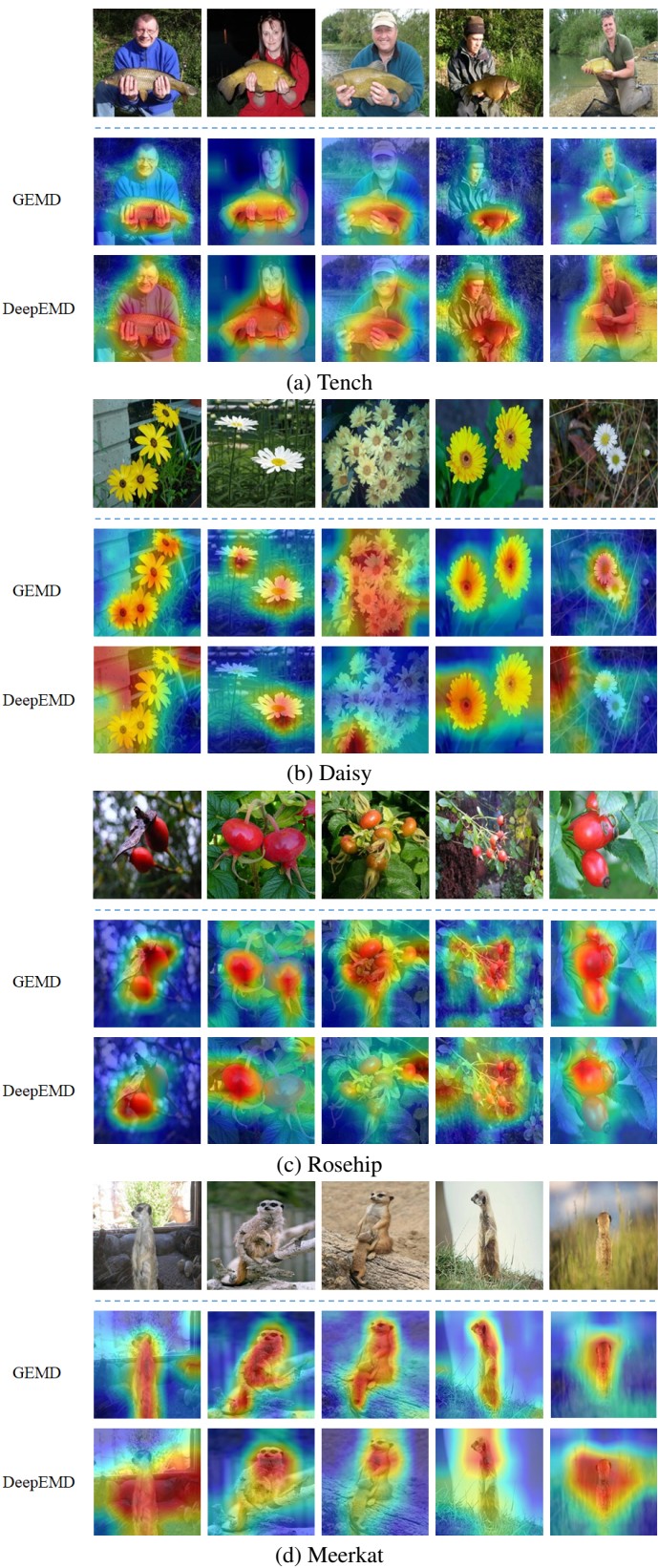

(a) Tench

(b) Daisy

(c) Rosehip

(d) Meerkat

Figure 3: Visualization through heatmaps with GEMD-T. Deeper color indicates more importance of the regions.

| Method | GEMD-M | GEMD-T |
|---|---|---|
| Baseline (frozen) | 77.0 | 78.7 |
| Naive finetuning | 75.6 | 77.4 |
| Our finetuning | **79.2** | **80.8** |

Table 5: 'Average All' accuracy (%) of different finetuning strategies with ResNet-18 on Meta-Dataset in MDL setting.

| Method | Dis-similarity | In-D | Out-D | All |
|---|---|---|---|---|
| GEMD-M | EMD | 62.0 | 72.8 | 72.0 |
| GEMD-T | EMD | 64.2 | 76.5 | 75.6 |
| ADM | KLD | 59.2 | 69.8 | 69.0 |
| Ours | G-JSD | 60.4 | 69.5 | 68.8 |
| Ours | Dual G-JSD | 60.0 | 70.0 | 69.2 |

Table 6: Comparison results (%) with Geometric Jensen-Shannon divergence with ResNet-18 on Meta-Dataset in SDL setting.

Our implementation of DeepEMD is based on the code repository released by the original authors. We implement the Sinkhorn algorithm via open source Python Optimal Transport (POT) library (Flamary et al., 2021). We use the code released by the authors of IPOT for its implementation. For each of the two solvers, we tune the regularizing parameter and iteration number on the validation splits. By careful tuning in terms of the balance between performance and speed, we set $\alpha = 0.03$ and the iteration number to 50 for the Sinkhorn solver, while choosing $\beta = 0.1$ and 40 iterations for the IPOT solver.

### A.4.2 IMPLEMENTATION OF RFS AND ADM

RFS (Tian et al., 2020) is closely related to our GEMD, since we adopt a similar methodology, i.e., pre-training with self-distillation technique on all classes of the training set and then train the model for every meta-testing task. We use implementation of the original authors that is public available. Note that we use a softmax classifier instead of the logistic regression model, as the former one runs significantly faster on GPU while achieving better performance.

ADM (Li et al., 2020) represents images with Gaussian descriptors and measures the dis-similarity with KL divergence. Its implementation is mainly based on the code released by the original authors. For a fair comparison, similar to our GEMD, we add one $1 \times 1$ convolution for dimension reduction before computing the mean vectors and covariance matrices. After tuning, we set its feature dimension to 512.

### A.5 ADDITIONAL EXPERIMENTS

In this section, we first present extra experiment to evaluate our simple finetuning strategy in MDL setting on Meta-Dataset. Next, we additionally compare with Jensen-Shannon divergence based FSL method. Finally, we give additional experiment on three small-scale benchmarks.

### A.5.1 OUR FINETUNING STRATEGY IN MDL SETTING ON META-DATASET

Recall that ResNet-18 consists of a $5 \times 5$ convolutional (conv) layer and 4 stages (i.e., conv2_x $\sim$ conv5_x). Every stage contains two residual blocks each of which is composed of two $3 \times 3$ conv layers and a shortcut connection, where each conv layer is followed by a batch normalization (BN) layer that is also learnable. The baseline method is concerned with freezing the backbone and training the classifier only, while the naive finetuning strategy simply unfreezes all learnable layers and trains the whole network. As regards our finetuning strategy, we only unfreeze the second conv layer of each residual block to prevent overfitting; meanwhile, we freeze BN layers since it is difficult to learn the statistics of novel data due to limited training examples.

Tab. 5 compares 'Average All' accuracy of different finetuning strategies. Compared to the baseline, the naive finetuning strategy is hurtful, decreasing the accuracy by $\sim 1.3\%$. In contrast, the proposed method improves the baseline by $\sim 2.1\%$, suggesting the effectiveness of our finetuning strategy. Nevertheless, it may be more beneficial to tweak the backone by designing domain- or task-specific adapters (Bateni et al., 2020; Dvornik et al., 2020; Li et al., 2022). Our GEMD methods focus on classifier at the network end, which is parallel to the adaptor-based methods and may further improve if combined with them. We leave this for future work.

| | *mini*ImageNet | *tiered*ImageNet | CUB | | Method | Hyper-parameter | *mini*ImageNet | *tiered*ImageNet | CUB |
|---|---|---|---|---|---|---|---|---|---|
| | (Vinyals et al.) | (Ren et al.) | (Chen et al.) | | | | (Vinyals et al.) | (Ren et al.) | (Chen et al.) |
| Image num | 60K | 779K | 12K | | | Initial LR | 1e-4 | 5e-5 | 1e-3 |
| Train | 64 | 351 | 100 | | GEMD-M | LR scheduler | [40, 80]*0.1 | [70]*0.1 | [40]*0.1 |
| Class Val | 16 | 97 | 50 | | | Epochs | 100 | 100 | 60 |
| Test | 20 | 160 | 50 | | | Initial LR | 5e-2 | 5e-2 | 5e-2 |
| Input size | 84×84 | 84×84 | 224×224 | | GEMD-T | LR scheduler | [100,150]*0.1 | [40,70]*0.1 | [300,350]*0.1 |
| Tasks | | **5-way 1-shot/5-shot** | | | | Epochs | 170 | 100 | 400 |

| (a) Statistics | (b) Hyper-parameters |
|---|---|

Table 7: Statistics of small-scale datasets and the hyper-parameter setting.

### A.5.2 COMPARISON WITH GEOMETRIC JENSEN-SHANNON DIVERGENCE IN SDL ON META-DATASET

In addition to KL divergence (KLD) based ADM, we would also like to compare with Jensen-Shannon divergence (JSD) (Lin, 1991), a well-known symmetric divergence. Unfortunately, when used to measure the discrepancy between two Gaussians, JSD has no closed from expression (Nielsen, 2019; Abou-Moustafa & Ferrie, 2012). Hence, we compare with geometric Jensen-Shannon divergence (G-JSD) recently proposed in Nielsen (2019) that can be computed in closed form.

Recall that $\mathcal{N}_X \triangleq \mathcal{N}(\boldsymbol{\mu}_X, \boldsymbol{\Sigma}_X)$ is a distribution with mean vector $\boldsymbol{\mu}_X$ and covariance matrix $\boldsymbol{\Sigma}_X$. The G-JSD between two Gaussians $\mathcal{N}_X$ and $\mathcal{N}_Y$ is defined as $d_{\text{G-JSD}} = (1 - \alpha)\text{KL}(\mathcal{N}_X\|\mathcal{N}_\alpha) + \alpha\text{KL}(\mathcal{N}_Y\|\mathcal{N}_\alpha)$ where $\alpha \in (0, 1)$ and $\text{KL}(\cdot\|\cdot)$ denotes KL divergence. Here $\mathcal{N}_\alpha$ is a Gaussian with mean vector $\boldsymbol{\mu}_\alpha$ and covariance matrix $\boldsymbol{\Sigma}_\alpha$, which take the forms $\boldsymbol{\mu}_\alpha = \boldsymbol{\Sigma}_\alpha\left((1-\alpha)\boldsymbol{\Sigma}_X^{-1}\boldsymbol{\mu}_X + \alpha\boldsymbol{\Sigma}_Y^{-1}\boldsymbol{\mu}_Y\right)$ and $\boldsymbol{\Sigma}_\alpha = \left((1-\alpha)\boldsymbol{\Sigma}_X^{-1} + \alpha\boldsymbol{\Sigma}_Y^{-1}\right)^{-1}$, respectively. We also compare with dual G-JSD for which $d_{\text{G-JSD}}^* = (1-\alpha)\text{KL}(\mathcal{N}_\alpha\|\mathcal{N}_X) + \alpha\text{KL}(\mathcal{N}_\alpha\|\mathcal{N}_Y)$.

We implement G-JSD and dual G-JSD in the framework of ADM as in Section A.4.2. After grid search for the parameter $\alpha$, we set it to 0.9 for the best performance. From Tab. 6, we can see that G-JSD and dual G-JSD achieve comparable accuracies, both perform on par with ADM. Compared to our GEMD, they significantly underperform by large margins. We hypothesize the reason is that KL-divergence based measures are agnostic to the metric of the underlying data distribution (Ozair et al., 2019); in contrast, our GEMD concerns a geodesic distance on the space of Gaussians (Malago et al., 2018).

### A.5.3 COMPARISON TO SOTA METHODS ON SMALL-SCALE DATASETS

**Datasets** To further evaluate GEMD, we conduct experiments on commonly used small-scale datasets, including *mini*ImageNet (Vinyals et al., 2016), *tired*ImageNet (Ren et al., 2018) and CUB-200-2011 (CUB) (Wah et al., 2011). The statistics of the three benchmarks are summarized in Tab. 7a.

*mini*ImageNet is a few-shot benchmark constructed from ImageNet (Deng et al., 2009) for general object recognition. It contains 100 categories each with 600 images, and we use standard splits provided in (Ravi & Larochelle, 2017), i.e., 64/16/20 classes for meta-train/val/test. For a fair comparison, we use ResNet-12 as the backbone and image resolution is $84\times84$.

*tiered*ImageNet is a few-shot benchmark also originated from ImageNet (Deng et al., 2009). Compared to *mini*ImageNet, it is larger and considers the hierarchical structure of ImageNet. Specifically, it consists of 608 classes from 34 super-classes and has 779,165 images in total, in which 20 super-classes (351 classes), 6 super-classes (97 classes) and 8 super-classes (160 classes) are used for meta-train, meta-val and meta-test, respectively.

CUB-200-2011 is a fine-grained benchmark, containing 200 bird classes with 11,788 images in total. We adopt the splits of Chen et al. (2019), where the overall classes are divided into 100/50/50 for meta-training/val/test. Following (Chen et al., 2019; Liu et al., 2020; Afrasiyabi et al., 2020), we make experiments with the original raw images.

**Settings**  Following the previous arts (Chen et al., 2021; Xie et al., 2022; Zhang et al., 2020), we pre-train the model on the overall categories of the training set. We perform meta-training for the metric-based GEMD-M; for each epoch, we randomly choose 1,000 5-way 1-shot episodes or 600 5-way 5-shot ones with 16 query images for each class. We do not perform meta-training for GEMD-T. During meta-testing, we randomly select 10,000 episodes for both 5-way 1-shot and 5-shot settings with 15 query images for each class; for every episode, we directly make inference without training for GEMD-M, while we train the classifier only with backbone frozen and then make inference for GEMD-T. As in (Zhang et al., 2020; 2022), we use standard data augmentation including random resized crop, random horizontal clip and color jittering. Across the experiments here, we adopt the SGD optimizer with a momentum of 0.9 and a weight decay of 0.0005. We summarize other hyper-parameters in Tab. 7b.

**Results**  As shown in Tab. 8a, for *general object recognition* our GEMD-M achieves competitive results, while our GEMD-T improves significantly over GEMD-M. This suggests again transfer learning-based method has better generalization capability than the metric-based method, which is consistent with our observation in the main paper. On *mini*ImageNet, compared to state-of-the-art methods, GEMD-T is the second best, only inferior to tSF for 1-shot tasks; it performs much better than all the competing methods for 5-shot tasks. On *tired*ImageNet, our GEMD-T achieves the best results for both 1-shot and 5-shot tasks. For *fine-grained recognition*, as Tab. 8b shows, both our GEMD-M and GEMD-T outperforms across the board. As the fine-grained categories pose very small inter-class difference but large intra-class variation, the results suggest our methods have the capability to learn more discriminative features. In summary, the experimental results in Tab. 8 show our methods are very competitive on the commonly used small-scale FSL datasets.

A.6  PROOF OF PROPOSITION 2

For self-contained and coherence, this section proves Proposition 2 in Section 4.3. One may refer to Bhatia et al. (2019) for more related conclusions.

**Proof 1**  *In terms of Eq. (4) and Eq. (7), we know that the task is to prove*

$$\text{tr}(\mathbf{\Sigma}_X + \mathbf{\Sigma}_Y - 2(\mathbf{\Sigma}_X\mathbf{\Sigma}_Y)^{\frac{1}{2}}) = \min_{\mathbf{U}} \parallel \mathbf{\Sigma}_X^{\frac{1}{2}}\mathbf{U} - \mathbf{\Sigma}_Y^{\frac{1}{2}} \parallel_F^2 . \tag{11}$$

*Consider a more general case where $\mathbf{U}$ is an unitary matrix. According to relation between Frobenious norm and matrix trace, Eq. (11) can be written as*

$$2\text{tr}((\mathbf{\Sigma}_X\mathbf{\Sigma}_Y)^{\frac{1}{2}}) = \max_{\mathbf{U}} \ \text{tr}(\mathbf{U}^*\mathbf{\Sigma}_X^{\frac{1}{2}}\mathbf{\Sigma}_Y^{\frac{1}{2}} + \mathbf{\Sigma}_Y^{\frac{1}{2}}\mathbf{\Sigma}_X^{\frac{1}{2}}\mathbf{U}), \tag{12}$$

*where $*$ denotes matrix conjugate transpose.*

*Let $\mathbf{Z} = \mathbf{\Sigma}_Y^{\frac{1}{2}}\mathbf{\Sigma}_X^{\frac{1}{2}}$ and $\mathbf{Z} = \mathbf{JS}$ be its polar decomposition, where $\mathbf{J}$ is a a unitary matrix and $\mathbf{S}$ is a positive semi-definite Hermitian matrix. It is not hard to show that $\mathbf{S} = (\mathbf{Z}^*\mathbf{Z})^{\frac{1}{2}} = (\mathbf{\Sigma}_X^{\frac{1}{2}}\mathbf{\Sigma}_Y\mathbf{\Sigma}_X^{\frac{1}{2}})^{\frac{1}{2}}$, and $\text{tr}(\mathbf{U}^*\mathbf{\Sigma}_X^{\frac{1}{2}}\mathbf{\Sigma}_Y^{\frac{1}{2}} + \mathbf{\Sigma}_Y^{\frac{1}{2}}\mathbf{\Sigma}_X^{\frac{1}{2}}\mathbf{U}) = \text{tr}(\mathbf{U}^*\mathbf{SJ}^* + \mathbf{JSU}) = \text{tr}(\mathbf{J}^*\mathbf{U}^*\mathbf{S} + \mathbf{UJS})$. Let $\mathbf{Q} = \mathbf{UJ}$ for which we know $\mathbf{QQ}^* = \mathbf{I}$. Then our task becomes how to prove*

$$2\text{tr}((\mathbf{\Sigma}_X\mathbf{\Sigma}_Y)^{\frac{1}{2}}) = \max_{\mathbf{U}} \text{tr}((\mathbf{Q} + \mathbf{Q}^*)\mathbf{S}). \tag{13}$$

*Let us choose a set of basis $\mathbf{Q} = \text{diag}(e^{i\theta_1}, \cdots, e^{i\theta_n})$ where diag denotes diagonal matrix. Then we have $\text{tr}((\mathbf{Q} + \mathbf{Q}^*)\mathbf{S}) = \sum_j^p 2\cos(\theta_j)\text{s}_{jj}$ where $s_{jj}$ denotes the diagonal entry of $\mathbf{S}$. Obviously, when $\cos(\theta_j) = 1$ for all $j$, i.e., $\mathbf{Q} = \mathbf{I}$, $\text{tr}((\mathbf{Q} + \mathbf{Q}^*)\mathbf{S})$ takes the maximum value $\sum_{j=1}^p 2\text{s}_{jj} = 2\text{tr}(\mathbf{S}) = 2\text{tr}((\mathbf{\Sigma}_X^{\frac{1}{2}}\mathbf{\Sigma}_Y\mathbf{\Sigma}_X^{\frac{1}{2}})^{\frac{1}{2}})$. As of now, we have*

$$\max_{\mathbf{U}} \text{tr}((\mathbf{Q} + \mathbf{Q}^*)\mathbf{S}) = 2\text{tr}((\mathbf{\Sigma}_X^{\frac{1}{2}}\mathbf{\Sigma}_Y\mathbf{\Sigma}_X^{\frac{1}{2}})^{\frac{1}{2}}). \tag{14}$$

*Note that when $\mathbf{Q} = \mathbf{UJ}$ is the identity matrix, we have $\mathbf{U} = \mathbf{J}^*$ which is the factorizer of the polar decomposition of $\mathbf{Z}^* = \mathbf{\Sigma}_X^{\frac{1}{2}}\mathbf{\Sigma}_Y^{\frac{1}{2}}$.*

| Method | Backbone | *mini*ImageNet | | *tiered*ImageNet | |
|---|---|---|---|---|---|
| | | 1-shot | 5-shot | 1-shot | 5-shot |
| DN4 (Li et al., 2019) | ResNet-12 | $64.73 \pm 0.44$ | $79.85 \pm 0.31$ | – | – |
| RFS (Tian et al., 2020) | ResNet-12 | $64.82 \pm 0.60$ | $82.14 \pm 0.43$ | $71.52 \pm 0.69$ | $86.03 \pm 0.58$ |
| FEAT (Ye et al., 2020) | ResNet-12 | $66.78 \pm 0.20$ | $82.05 \pm 0.14$ | $70.80 \pm 0.23$ | $84.79 \pm 0.16$ |
| Meta-Baseline (Chen et al., 2021) | ResNet-12 | $63.17 \pm 0.23$ | $79.26 \pm 0.17$ | $68.62 \pm 0.27$ | $83.29 \pm 0.18$ |
| MELR (Fei et al., 2021) | ResNet-12 | $67.40 \pm 0.43$ | $83.40 \pm 0.28$ | $72.14 \pm 0.51$ | $87.01 \pm 0.35$ |
| FRN (Wertheimer et al., 2021) | ResNet-12 | $66.45 \pm 0.19$ | $82.83 \pm 0.13$ | $71.16 \pm 0.22$ | $86.01 \pm 0.15$ |
| IEPT (Zhang et al., 2021) | ResNet-12 | $67.05 \pm 0.44$ | $82.90 \pm 0.30$ | $72.24 \pm 0.50$ | $86.73 \pm 0.34$ |
| BML (Zhou et al., 2021) | ResNet-12 | $67.04 \pm 0.63$ | $83.63 \pm 0.29$ | $68.99 \pm 0.50$ | $85.49 \pm 0.34$ |
| ProtoNet (Snell et al., 2017) | ResNet-12 | $62.11 \pm 0.44$ | $80.77 \pm 0.30$ | $68.31 \pm 0.51$ | $83.85 \pm 0.36$ |
| ADM (Li et al., 2020) | ResNet-12 | $65.87 \pm 0.43$ | $82.05 \pm 0.29$ | $70.78 \pm 0.52$ | $85.70 \pm 0.43$ |
| CovNet (Wertheimer & Hariharan, 2019) | ResNet-12 | $64.59 \pm 0.45$ | $82.02 \pm 0.29$ | $69.75 \pm 0.52$ | $84.21 \pm 0.26$ |
| DeepEMD (Zhang et al., 2020) | ResNet-12 | $65.91 \pm 0.82$ | $82.41 \pm 0.56$ | $71.16 \pm 0.87$ | $86.03 \pm 0.58$ |
| UNICORN-MAML (Ye & Chao, 2022) | ResNet-12 | $65.17 \pm 0.20$ | $84.30 \pm 0.14$ | $69.24 \pm 0.20$ | $86.06 \pm 0.16$ |
| TAS-distill (Le et al., 2022) | ResNet-12 | $65.13 \pm 0.39$ | $82.47 \pm 0.52$ | $72.81 \pm 0.48$ | $87.21 \pm 0.52$ |
| SetFeat (Afrasiyabi et al., 2022) | SF-12 | $68.32 \pm 0.62$ | $82.71 \pm 0.46$ | $73.63 \pm 0.88$ | $87.59 \pm 0.57$ |
| MCL-Katz (Liu et al., 2022) | ResNet-12 | $67.51 \pm 0.20$ | $83.99 \pm 0.20$ | $72.01 \pm 0.20$ | $86.02 \pm 0.20$ |
| Meta DeepBDC (Xie et al., 2022) | ResNet-12 | $67.34 \pm 0.43$ | $84.46 \pm 0.28$ | $72.34 \pm 0.49$ | $87.31 \pm 0.32$ |
| tSF (Lai et al., 2022) | ResNet-12 | $\mathbf{69.74} \pm 0.47$ | $83.91 \pm 0.30$ | $71.98 \pm 0.50$ | $85.49 \pm 0.35$ |
| CORL (He et al., 2023) | ResNet-12 | $65.74 \pm 0.53$ | $83.03 \pm 0.33$ | $73.82 \pm 0.58$ | $86.76 \pm 0.52$ |
| LP-FT-FB (Wang et al., 2023) | ResNet-18 | 66.20 | 83.06 | 73.98 | 89.48 |
| GEMD-M (ours) | ResNet-12 | $66.30 \pm 0.20$ | $84.63 \pm 0.12$ | $71.71 \pm 0.22$ | $88.78 \pm 0.14$ |
| GEMD-T (ours) | ResNet-12 | $67.98 \pm 0.19$ | $\mathbf{85.09} \pm 0.12$ | $\mathbf{74.81} \pm 0.47$ | $\mathbf{89.93} \pm 0.32$ |

(a) General object recognition

| Method | Backbone | CUB | |
|---|---|---|---|
| | | 1-shot | 5-shot |
| MatchNet (Vinyals et al., 2016) | ResNet-12 | $71.87 \pm 0.85$ | $85.08 \pm 0.57$ |
| MAML (Finn et al., 2017) | ResNet-18 | $68.42 \pm 1.07$ | $83.47 \pm 0.62$ |
| $\Delta$-encoder (Schwartz et al., 2018) | ResNet-18 | 69.80 | 82.60 |
| Baseline++ (Chen et al., 2019) | ResNet-18 | $67.02 \pm 0.90$ | $83.58 \pm 0.54$ |
| AA (Afrasiyabi et al., 2020) | ResNet-18 | $74.22 \pm 1.09$ | $88.65 \pm 0.55$ |
| Neg-Cosine (Liu et al., 2020) | ResNet-18 | $72.66 \pm 0.85$ | $89.40 \pm 0.43$ |
| LaplacianShot (Ziko et al., 2020) | ResNet-18 | 80.96 | 88.68 |
| FRN (Wertheimer et al., 2021) | ResNet-18 | $82.55 \pm 0.19$ | $92.98 \pm 0.10$ |
| RFS (Tian et al., 2020) | ResNet-18 | $77.92 \pm 0.46$ | $89.94 \pm 0.26$ |
| ProtoNet (Snell et al., 2017) | ResNet-18 | $80.90 \pm 0.43$ | $89.81 \pm 0.23$ |
| ADM (Li et al., 2020) | ResNet-18 | $79.31 \pm 0.43$ | $90.69 \pm 0.21$ |
| CovNet (Wertheimer & Hariharan, 2019) | ResNet-18 | $80.76 \pm 0.42$ | $92.05 \pm 0.20$ |
| DeepEMD (Zhang et al., 2020) | ResNet-12 | $75.65 \pm 0.83$ | $88.69 \pm 0.50$ |
| UNICORN-MAML (Zhang et al., 2020) | ResNet-12 | $78.07 \pm 0.20$ | $91.67 \pm 0.16$ |
| ProtoNet(+OT) (Guo et al., 2022) | ResNet-10 | – | $84.44 \pm 0.51$ |
| SetFeat (Afrasiyabi et al., 2022) | SF-12 | $79.60 \pm 0.80$ | $90.48 \pm 0.44$ |
| Meta DeepBDC (Xie et al., 2022) | ResNet-18 | $83.55 \pm 0.40$ | $93.82 \pm 0.17$ |
| LP-FT-FB (Wang et al., 2023) | ResNet-18 | 73.36 | 86.88 |
| GEMD-M (ours) | ResNet-18 | $84.75 \pm 0.40$ | $94.25 \pm 0.28$ |
| GEMD-T (ours) | ResNet-18 | $\mathbf{84.95} \pm 0.43$ | $\mathbf{94.87} \pm 0.28$ |

(b) Fine-grained recognition

Table 8: Comparison on small-scale FSL benchmarks.

*We note that $\boldsymbol{\Sigma}_X \boldsymbol{\Sigma}_Y = \boldsymbol{\Sigma}_X^{\frac{1}{2}} (\boldsymbol{\Sigma}_X^{\frac{1}{2}} \boldsymbol{\Sigma}_Y \boldsymbol{\Sigma}_X^{\frac{1}{2}}) \boldsymbol{\Sigma}_X^{-\frac{1}{2}}$, whose square root takes the form $(\boldsymbol{\Sigma}_X \boldsymbol{\Sigma}_Y)^{\frac{1}{2}} = \boldsymbol{\Sigma}_X^{\frac{1}{2}} (\boldsymbol{\Sigma}_X^{\frac{1}{2}} \boldsymbol{\Sigma}_Y \boldsymbol{\Sigma}_X^{\frac{1}{2}})^{\frac{1}{2}} \boldsymbol{\Sigma}_X^{-\frac{1}{2}}$ owing to the property of matrix similarity. Therefore, we have*

$$\operatorname{tr}((\boldsymbol{\Sigma}_X \boldsymbol{\Sigma}_Y)^{\frac{1}{2}}) = \operatorname{tr}((\boldsymbol{\Sigma}_X^{\frac{1}{2}} \boldsymbol{\Sigma}_Y \boldsymbol{\Sigma}_X^{\frac{1}{2}})^{\frac{1}{2}}), \tag{15}$$

*where we use the property that the trace of a product is commutative. Combining Eq. (14) and Eq. (15), we can complete our proof.*

### A.7 LIMITATIONS AND FUTURE RESEARCH

In this paper, we assume deep features follow Gaussian distributions. Despite the facts that Gaussian is of maximal entropy for given mean and covariance and has closed form distance, this assumption may not be optimal or even do hold. Depending on various architectures of deep neural networks (DNNs), feature distributions may be multimodal (multiple peaks) and, if single-peaked, may be different functions other than Gaussian. As far as we know, distribution of deep features is inadequately studied and future research on this topic can potentially further benefit representation learning in the few-shot regime.

Our GEMD is not applicable to scenarios where distributions are discrete, e.g., categorical distribution or histogram; in these cases, DeepEMD is a natural option. In addition, GEMD only focuses on classifier attached at the end of DNNs without touching on middle layers; in contrast, recent state-of-the-art methods design task-/domain-specific adapters, i.e., auxiliary lightweight sub-networks, for tweaking the middle layers of the backbone. GEMD is parallel to such methods and it is promising to combine them for further improvement.

