# OpenReview forum: "Towards Faster and Stronger Deep Earth Mover's Distance for Few-Shot Learning"
_ICLR.cc/2024/Conference — ICLR 2024 Conference Withdrawn Submission_

### Official Review · Reviewer_YQkJ · 2023-10-26

**Soundness:** 3 good
**Presentation:** 3 good
**Contribution:** 3 good
**Rating:** 6
**Confidence:** 4

**Summary:**

The paper proposes a metric-based Gaussian Earth Mover's Distance (GEMD) for few-shot learning, which is computationally more efficient than the existing DeepEMD method. GEMD is implemented using transfer learning and a learnable metric, achieving superior performance compared to DeepEMD

**Strengths:**

The proposed GEMD method achieves compelling performance compared to state-of-the-art methods, as demonstrated through extensive experiments on large-scale Meta-Dataset and three small-scale benchmarks .

**Weaknesses:**

* Limited discussion on the limitations of GEMD: The paper does not provide a comprehensive discussion on the limitations of the proposed GEMD method. It would be beneficial to include a section discussing the potential drawbacks or scenarios where GEMD may not perform as well, providing insights for future research and potential improvements
* lack of visualization and analysis: The paper  does not provide any visualizations or in-depth analysis of the results. Including visualizations and further analysis of the learned features or matching flows would provide a deeper understanding of the proposed GEMD method and its effectiveness

**Questions:**

See weakness section for more details.

---

> ### Author Response · Authors · 2023-11-20
>
> We are grateful for the positive comments on the soundness, presentation and contribution of our manuscript. In particular, we appreciate the comments such as "compelling performance" and "extensive experiments". Hopefully, our answers could address your concerns.
>
> *Q1: Limited discussion on the limitations of GEMD: The paper does not provide a comprehensive discussion on the limitations of the proposed GEMD method. It would be beneficial to include a section discussing the potential drawbacks or scenarios where GEMD may not perform as well, providing insights for future research and potential improvements.*
>
> Our answer:
>
> Thanks for the comment. In the last paragraph of Section 6 titled "Conclusion and Limitations" of the original manuscript, we briefly introduce drawbacks of GEMD. **Following your suggestion, we add separately a new section titled "Limitations and Future Research", discussing more comprehensively the limitations of GEMD. Kindly refer to Section A.7 of the modified paper.** The discussion is also attached below for ease of reference.
>
> In this paper, we assume deep features follow Gaussian distributions. Despite the facts that Gaussian is of maximal entropy for given mean and covariance and has closed form distance, this assumption may not be optimal or even do hold. Depending on various architectures of deep neural networks (DNNs), feature distributions may be multimodal (multiple peaks) and, if single-peaked, may be different functions other than Gaussian. As far as we know, distribution of deep features is inadequately studied and future research on this topic can potentially further benefit representation learning in the few-shot regime.
>
> Our GEMD is not applicable to scenarios where distributions are  discrete, e.g., categorical distribution or  histogram; in these cases,  DeepEMD is a natural option. In addition, GEMD only focuses on classifier attached at the end of DNNs without touching on middle layers; in contrast, recent state-of-the-art methods design task-/domain-specific adapters, i.e., auxiliary lightweight sub-networks, for tweaking the middle layers of the backbone. GEMD is parallel to such methods and it is promising to combine them for further improvement.
>
> *Q2: Lack of visualization and analysis: The paper does not provide any visualizations or in-depth analysis of the results. Including visualizations and further analysis of the learned features or matching flows would provide a deeper understanding of the proposed GEMD method and its effectiveness.*
>
> Our answer:
>
> Thank you for the concern. In Section A.3 titled "Visualization of GEMD" of the original paper,  we visualize the effect of patch matching of GEMD-M and compare with DeepEMD in 1-shot setting. Specifically, we provide visualization of matching of typical matches for four categories, i.e., mushroom, arctic fox, school bus and beer bottle. In most cases, GEMD shows better feature matchings than DeepEMD in few-shot learning. The visualization results show that the matching flows determined by GEMD have better capability to discriminate between the query and support images. Kindly see Section A.3 for details.
>
> **In terms of your suggestion, in Section A.3 of the modified paper, we add additional analysis for visualizing the transfer learning based GEMD-T through commonly used heatmaps.**
>
> For ease of reference, we also give brief discussion on visualization by heatmaps in the following; kindly refer to Section A.3 for complete visualization results and discussion.
>
> We visualize the heapmaps of images by using CAM (Zhou et al. ,2016). We select four categories, i.e., Tench, Daisy, Rosehip and Meekat, for visualization with models learned by GEMD-T and DeepEMD.  On the whole, our GEMD can accurately focus on the regions of the objects while neglecting the backgrounds; in contrast, DeepEMD has larger tendency to be distracted by backgrounds. The comparison suggests that GEMD can learn more discriminative features than DeepEMD, demonstrating better capability of representation learning.

---

### Official Review · Reviewer_dGAQ · 2023-10-28

**Soundness:** 2 fair
**Presentation:** 2 fair
**Contribution:** 2 fair
**Rating:** 5
**Confidence:** 4

**Summary:**

The authors add the Gaussian EMD metric to the few-shot learning by modeling each channel feature (local feature) of each image as a Gaussian model, and then the EMD between Gaussian models is used as the metric of ProtoNet. Meanwhile, the authors accelerate the computational process by parameterizing the EMD metric so that it can be trained and computed on GPUs, enabling it to cover large-scale few-sample learning datasets.

**Strengths:**

In few-shot learning, it is important to explicitly represent the local features of the samples, and how to define the local features is very difficult to achieve in complex image distributions. By extracting and modeling the features of each channel of an image into a Gaussian model, the authors can model the local features without prior knowledge and match the local features between different samples by Gaussian EMD, which provides a new way of thinking for the representation of local features in few-shot learning.
The authors' experiments are adequate on both large-scale Meta-dataset and small-scale datasets to illustrate the validity of the method.

**Weaknesses:**

This work is lack of novelty. The essence of this work is to use Gaussian EMD as a metric based on ProtoNet, which is still not novel enough, although some existing methods are used to accelerate the process.
The figures in the paper are somewhat obscure, such as Fig 1(b). There are some typos in the writing, such as "Wang et al. (2017) propose" and the following "Bilinear pooling (Lin et al., 2018) or covariance pooling (Wang et al., 2021; Song et al., 2023) yields" in p3.

**Questions:**

The authors describe that the method consists of two stages, pre-training and meta-test. Please elaborate what is the loss in meta-test and if possible how to finetune the network by using the loss calculated based on Gaussian EMD metric.
In the experiments, the authors used the pre-train model of Resnet-18 and the self-distillation, while the effect of the base model (the above two things) on the performance of the Meta-dataset is still vague. Please give descriptions of the comparison methods or mark in the tables, the difference between the comparison method and GEMD on the base model.
There are a few grammatical errors in the text that need to be fixed, such as "Wang et al. (2017) propose" and the following "Bilinear pooling (Lin et al., 2018) or covariance pooling (Wang et al., 2021; Song et al., 2023) yields" in p3， "in SDL setting we freeze the backbone networks that are used to…" in p7, etc.

---

> ### Author Response · Authors · 2023-11-21
>
> Thanks for the constructive comments. In particular, we appreciate the positive comments, e.g., "a new way of thinking for the representation of local features in few-shot learning", and "experiments are adequate".
>
> *Q1: This work is lack of novelty. The essence of this work is to use Gaussian EMD as a metric based on ProtoNet, which is still not novel enough, although some existing methods are used to accelerate the process.*
>
> Our answer:
>
> We are afraid there is a misunderstanding on the novelty of our manuscript.
>
> Above all, our GEMD-T is transfer learning based method, ***NOT based on ProtoNet***. GEMD-T proposes to learn a parametric EMD with learnable orthogonal matrices for a more discriminative Gaussian metric. As far as we know, this is ***the first attempt*** that enables the EMD metric for Gaussians to be learned parametrically in deep learning.
>
> Secondly, inspired by DeepEMD, our GEMD-M proposes Gaussian for modeling distribution and closed form EMD for FSL. We illuminate that Gaussian EMD amounts to feature matching and the optimal flows follow a joint Gaussian, which ***has not been elucidated previously*** in deep learning to our best knowledge, providing an intuitive interpretation of the metric.
>
> *Q2: The authors describe that the method consists of two stages, pre-training and meta-test. Please elaborate what is the loss in meta-test and if possible how to finetune the network by using the loss calculated based on Gaussian EMD metric.*
>
> Our answer:
>
> Thanks for the concern.
>
> In the setting of SDL, for each episode, we freeze the backbone of the pre-trained model and train the classifier only. For the metric-based GEMD-M with loss function in Eq. (6), we initialize the nearest neighbor classifier with the prototypical Gaussians of support classes and then trained in 5 epochs with a learning rate of 0.001, a weight decay of 0.0001 and a batch size of 64. For the transfer learning based GEMD-T with loss function in Eq. (8), after freezing the backbone of the pre-trained model, the softmax-like classifier is trained from scratch for 600 iterations with a learning rate of 0.01 and a weight decay 0.0005.
>
> In MDL, We finetune the model with a weight decay of 0.0001, in which we only unfreeze the second conv layer of every residual block to avoid the risk of overfitting. For GEMD-M with loss function in Eq. (6), after initializing the classifier with the prototypes of support classes, we finetune the network for 50 epochs with a learning rate of 0.0001 and a batch size of 32. For GEMD-T with loss function in Eq. (8), we adopt a dropout (rate=0.7) for the last FC layer; the network is finetuned for 70 epochs with the learning rate set to 0.005.
>
> **Kindly note that, in Section A.2 of Appendix, we elaborated the two-stage pipeline (i.e., pre-training and meta-testing), and the implementation details including how to finetune the network for both SDL and MDL. See Section A.2 for more details.**
>
>
> *Q3: In the experiments, the authors used the pre-train model of Resnet-18 and the self-distillation, while the effect of the base model (the above two things) on the performance of the Meta-dataset is still vague. Please give descriptions of the comparison methods or mark in the tables, the difference between the comparison method and GEMD on the base model.*
>
> Our answer:
>
> State of the art methods, including URL, TSA and 2LM+TSA, adopt the same base model in light of the pipeline of URL, i.e., pretraining multiple models and then training one single model via teacher-student distillation. Specifically, they pre-train a network for each dataset and the resulting  multiple pre-trained models are then used as teacher to distill a universal student model. Differently, our GEMD follows the pipeline of RFS, i.e., pretraining a single network on all categories concatenated together from all datasets followed by self-distillation. For fair comparison, we implement our counterparts including DeepEMD and ADM with the same pipeline as RFS. Notably, in SLD setting where there is only one training dataset, the pipeline of URL boils down to that of RFS, so establishing interesting connection between them. Note that the pipeline of RFS is simpler than that of URL. The performance of GEMD can potentially improve further if we adopted the URL’s pipeline.
>
> *Q4: The figures in the paper are somewhat obscure, such as Fig 1(b). There are a few grammatical errors in the text that need to be fixed, such as "Wang et al. (2017) propose" and the following "Bilinear pooling (Lin et al., 2018) or covariance pooling (Wang et al., 2021; Song et al., 2023) yields" in p3， "in SDL setting we freeze the backbone networks that are used to…" in p7, etc.*
>
> Our answer:
>
> Thanks. We will try best to polish the presentation of our manuscript.

---

> > ### Comment · Reviewer_dGAQ · 2023-11-22
> >
> > Thank the authors for the response, and I have read via the feedback. I maintain my review score.

---

> > > ### Author Response · Authors · 2023-11-22
> > >
> > > Thanks for the feedback. If you elaborated your comments on our rebuttal, we would be able to provide further clarification.

---

### Official Review · Reviewer_rDfm · 2023-11-01

**Soundness:** 3 good
**Presentation:** 2 fair
**Contribution:** 3 good
**Rating:** 6
**Confidence:** 4

**Summary:**

In this work, the authors propose two methods, GEMD-M and GEMD-T, built upon DeepEMD and utilizing Gaussian distribution for modeling. GEMD-M method demonstrates that EMD can be computed in a closed-form and formulates EMD as a feature-matching problem where features follow a joint Gaussian distribution. However, the EMD computation in GEMD-M is entangled and unsuited for GPU calculations. In contrast, GEMD-T introduces learnable orthogonal matrices to achieve parameterized learning of EMD and resolves the entanglement in calculations. Experimental results show that the GEMD method outperforms existing methods on cross-domain few-shot datasets such as Meta-dataset.

**Strengths:**

This paper introduces an outstanding few-shot recognition method, demonstrating the effectiveness of GEMD. By ingeniously employing Gaussian distribution modeling, it successfully addresses the high computational cost issue in DeepEMD. The paper introduces the GEMD method with learnable parameters, achieving decoupling of computations and further enhancing computational speed.

**Weaknesses:**

1. Several analyses in the paper lack experimental validation, as follows:
1) The paper mentions the drawbacks of KL divergence but lacks experimental results in the ablation studies to demonstrate whether GEMD outperforms KL divergence. This necessitates further experimental evidence from the authors.s
2) The paper mentions in both the title and the main text that GEMD-M and GEMD-T contribute to computational speed but lack experimental results to verify their effectiveness.
2. The conclusions in the paper are directly borrowed from the conclusions of previous articles, especially GEMD-T, where the derivation of some essential steps is missing, leading to a lack of coherence.

**Questions:**

1.Why wasn't the matching cost 1-cos(x, y) from DeepEMD used, and L2 distance was chosen instead? Does using 1-cos(x, y) still yield a more concise expression for EMD?
2. Can EMD be employed for the distillation learning of teacher-student models? If so, should using EMD for result distillation be considered in URL?"

---

> ### Author Response · Authors · 2023-11-20
>
> We are thankful for your positive feedback on our paper. In particular, we appreciate your comments, e.g., “an outstanding few-shot recognition method” and “ingeniously employing Gaussian distribution modeling”. We hope our answers could address your concerns.
>
> *Q1: The paper mentions the drawbacks of KL divergence but lacks experimental results in the ablation studies to demonstrate whether GEMD outperforms KL divergence. This necessitates further experimental evidence from the authors.*
>
> Our answer:
>
> Kindly note ADM (Li et al., 2020) is a KL divergence (KLD) based FSL method with Gaussian descriptor, which is compared in Tables. 1 and 2 (Meta-Dataset) and Table 3 (small-scale datasets).
>
> For ease of reference, we also present the results of KLD based ADM vs. GEMD below. These comparisons clearly show superiority of EMD over KLD as dis-similarity measure of Gaussian descriptors.
>
> Comparison on Meta-Dataset in SDL and MDL settings
>
> |Method|Dis_similarity|In-D (SDL)|Out-D (SDL)| All (SDL)$\ \ \ \ $$\mid$$\ \ \ \$|In-D (MDL)|Out-D (MDL)|All (MDL)|
> |:-:|:-:|:-:|:-:|:-:|:-:|:-:|:-:|
> | ADM|KLD|59.2|69.8|69.0$\ \ \ \ \ \ \ \ \$$\mid$|81.2|73.0|78.1|
> |GEMD-M|EMD|62.0|72.8|72.0$\ \ \ \ \ \ \ \ \$$\mid$|81.0|76.3|79.2|
> |GEMD-T|EMD|64.2|76.5|75.6$\ \ \ \ \ \ \ \ \$$\mid$|82.5|78.0|80.8|
>
> Comparison on MiniImageNet(miniIN), TieredImageNet (TieredIN) and CUB
>
> |Method$\ \ \ $|$\ \ \ \ \ \ \ \ \ \ $Dis_similarity$\ \ \ \ \ \ \ \ \ \ \ \ \ \ \ \ \ \ \ \ \ \ $|MiniIN 1shot|MiniIN$\ \ \ \ \ \ \ \ \ \ $5shot$\ \ \ \ \ \ \ \ \ \ \ $$\mid$|TieredIN 1shot|TieredIN$\ \ \ \ \ \ \ \ \ \ \ \ \ \ \ \ \$5shot$\ \ \ \ \ \ \ \ \ \ \ \ $$\mid$|CUB 1shot|CUB 5shot|
> |:-:|:-:|:-|:-|:-|:-|:-|:-|
> |ADM|KL divergence|65.87|82.05$\ \ \ \ \ \ \ \ \ \ \ \ \$$\mid$|70.78|85.70$\ \ \ \ \ \ \ \ \ \ \ \ \ $$\mid$|79.31|90.69|
> |GEMD-M|Earth Mover’s Distance|66.30|84.63$\ \ \ \ \ \ \ \ \ \ \ \ \$$\mid$|71.71|88.78$\ \ \ \ \ \ \ \ \ \ \ \ \ $$\mid$|84.75|94.25|
> |GEMD-T|Earth Mover’s Distance|67.98|85.09$\ \ \ \ \ \ \ \ \ \ \ \ $$\mid$|74.81|89.93$\ \ \ \ \ \ \ \ \ \ \ \ \ $$\mid$|84.95|94.87|
>
> **Additionally, we add comparison with Geometric Jensen-Shannon divergence (G-JSD) in the modified paper. Kindly see Section A.5.2 for details on formulas, implementations and results.**
>
> *Q2: The paper mentions in both the title and the main text that GEMD-M and GEMD-T contribute to computational speed but lack experimental results to verify their effectiveness.*
>
> Our answer:
>
> Kindly note we compare wall-clock time between DeepEMD and GEMD in Table 3e. Besides, Section A.2.3 in Appendix gives complexity analysis.
>
> For ease of reference, we also present the wall-clock time below. It can be seen GEMD-M improves the speed of DeepEMD while having much better performance; GEMD-T is significantly faster than DeepEMD and meanwhile outperforms it by a large margin. Our GEMD-T is clearly superior to RFS while having almost the same speed.
>
> | Method | Acc (%) |   Time (ms)   |
> |:------:|:--------:|:---------:|
> |  DeepEMD-QPTH  |    -   |  >700K     |
> |  DeepEMD-Sinkhorn    |    69.5 |  420     |
> |  DeepEMD-IPOT     |    69.4   |  540    |
> |  GEMD-M  |    72.0   |  328     |
> |  GEMD-T    |   75.6 |  64     |
> |  RFS     |    68.8  |  60    |
>
> **Additionally, we add wall-clock time comparison with state-of-the-art TSA and ProtoNet in the modified manuscript (cf. Table 3e).**
>
> *Q3: The conclusions in the paper are directly borrowed from the conclusions of previous articles, especially GEMD-T, where the derivation of some essential steps is missing, leading to a lack of coherence.*
>
> Our answer:
>
> Thanks for the concern. **In terms of your suggestion, for GEMD-T, we add separately Section A.6, providing essential steps for the proof. Kindly see the modified manuscript for details.**
>
>
> *Q4: Why wasn't the matching cost 1-cos(x, y) from DeepEMD used, and L2 distance was chosen instead? Does using 1-cos(x, y) still yield a more concise expression for EMD?*
>
> Our answer:
>
> As far as we know, except L2 distance, for other distances such as L1, Lp (p>2) and 1-cos(x,y), Gaussian EMD has no closed form expression. This is the reason why we choose L2 distance. EMD between Gaussians with 1-cos(x,y) as the cost may improve L2 distance, whose solution, however, is an open problem that requires theoretical investigation.
>
>
> *Q5: Can EMD be employed for the distillation learning of teacher-student models? If so, should using EMD for result distillation be considered in URL?*
>
> Our answer:
>
> Thanks for the questions. EMD in discrete form can handle categorical distribution, so can potentially be used in prediction (logits) based distillation learning, instead of the classical KLD. Gaussian EMD (GEMD) may be used in intermediate layers for knowledge transfer from features. In the framework of URL where a universal model is distilled from separately trained multiple models, EMD and GEMD can potentially replace KLD and CKA for transferring multi-domain knowledge, respectively.

---

### Official Review · Reviewer_epv6 · 2023-11-12

**Soundness:** 3 good
**Presentation:** 2 fair
**Contribution:** 2 fair
**Rating:** 6
**Confidence:** 3

**Summary:**

This paper mainly targets few-shot learning. The authors concentrate on the previous metric-based method DeepEMD. Sepcifically, they point out the drawbacks of DeepEMD in terms of efficiency, and accordingly propose to leverage the Gaussian EMD metric to replace the discrete EMD version. The authors further propose two instantiations for better parallel on GPU devices. The proposed method is evaluated on several datasets including Meta-Dataset, miniImageNet, tieredImageNet and CUB to show the effectiveness.

**Strengths:**

1. The idea of using Gaussian EMD is solid and interesting for few-shot learning.
2. The authors have provided extensive experiment results.

**Weaknesses:**

1. As explained in Sec.4.2, the method averages statistics from different samples to estimate the prototype. I wonder the difference between such implementation and treating all local descriptors from the same class as random samples from one Gaussian distribution and then estimating the statistics.

2. The authors mention in the paper that KL divergence is not suitable because it is asymmetric. What about Jensen-Shannon divergence?

3. As for the efficiency comparison in Tab.3e, I think it would be better to compare the proposed method with not only EMD, but also other methods using pretrain and meta-train pipeline.

4. I notice that the proposed method performs worse than recent finetuning-based method [1,2] in SDL setting.

[1] Xu C, Yang S, Wang Y, et al. Exploring efficient few-shot adaptation for vision transformers. TMLR 2022.

[2] Basu S, Massiceti D, Hu S X, et al. Strong Baselines for Parameter Efficient Few-Shot Fine-tuning. arXiv 2023.

**Questions:**

Please refer to the weaknesses.

---

> ### Author Response · Authors · 2023-11-20
>
> Thanks for the insightful and constructive comments. Particularly, we are thankful for the positive comments for acknowledging that our idea is solid and interesting and that our experiment results are extensive.
>
> We would like to emphasize that, besides metric based GEMD-M, we propose transfer learning based GEMD-T, which, ***for the first time*** to our best knowledge, learns a parametric EMD metric with learnable orthogonal matrices for Gaussians in deep learning.
>
> Hopefully, our point-to-point answer could address your concerns.
>
> *Q1: As explained in Sec.4.2, the method averages statistics from different samples to estimate the prototype. I wonder the difference between such implementation and treating all local descriptors from the same class as random samples from one Gaussian distribution and then estimating the statistics.*
>
> Our answer:
>
> Thanks. We implement the strategy you suggested to compute the prototypical Gaussian, called Strategy B for convenience. We compare it with our original method (Strategy A) in SDL setting on Meta-Dataset. Strategy B underperforms Strategy A for In-Domain (In-D) tasks while outperforming for Out-of-Domain tasks (Out-D). Overall, they perform comparatively.
>
> |Strategy | Method |In-D | Out-D | All |
> |:--------:|:--------:|:---------:|:---------:|:----------:|
> |A|  Average samples  |  73.2    | 70.0    | 72.0    |
> |B| All local descriptors  |  72.1    | 70.9    | 71.6    |
>
> *Q2: The authors mention in the paper that KL divergence is not suitable because it is asymmetric. What about Jensen-Shannon divergence?*
>
> Our answer:
>
> Note that Jensen-Shannon divergence (JSD) between two Gaussians has no closed from expression (Nielsen, 2019). Hence, we compare with geometric Jessen-Shannon divergence (G-JSD) and dual G-JSD recently proposed in Nielsen (2019) that can be computed in closed form. G-JSD and dual G-JSD achieve comparable accuracies, both perform on par with ADM. Compared to our GEMD, they significantly underperform by large margins.
>
> |Method|Dis-similarity|In-D|Out-D|All|
> |:-|:-|:-:|:-:|:-:|
> | GEMD-M | EMD | 62.0 | 72.8 | 72.0 |
> | GEMD-T | EMD | 64.2 | 76.5 | 75.6 |
> | ADM | KLD | 59.2 | 69.8 | 69.0 |
> | Ours | G-JSD | 60.4 | 69.5 | 68.8 |
> | Ours | Dual G-JSD | 60.0 | 70.0 | 69.2 |
>
> **Details on theory, implementation and discussion of (dual) G-JSD are given in the modified paper (cf. Section A.5.2).**
>
> *Q3: As for the efficiency comparison in Tab.3e, I think it would be better to compare the proposed method with not only EMD, but also other methods using pretrain and meta-train pipeline.*
>
> Our answer:
>
> Following your suggestion, we compare with state-of-the-art TSA, together with a commonly used baseline of ProtoNet. Compared to TSA, our GEMD-T is 2 times faster and meanwhile performing better.  ProtoNet consists of  pre-training plus meta-training along with meta-test where no training is required. Hence, ProtoNet runs the fastest, but with significantly inferior performance.
>
> | Method | Acc (%) |   Time (ms)   |
> |:------:|:--------:|:---------:|
> |  DeepEMD-QPTH  |    -   |  >700K     |
> |  DeepEMD-Sinkhorn    |    69.5 |  420     |
> |  DeepEMD-IPOT     |    69.4   |  540    |
> |  GEMD-M  |    72.0   |  328     |
> |  GEMD-T    |   75.6 |  64     |
> |  RFS     |    68.8  |  60    |
> |  TSA     |    73.3  |  120    |
> |  ProtoNet     |   56.1  |  45    |
>
> **Comparisons with TSA and ProtoNet are incorporated in the modified paper (cf. Table 3e).**
>
>
> *Q4: I notice that the proposed method performs worse than recent finetuning-based method [1,2] in SDL setting.
> [1] Xu C, Yang S, Wang Y, et al. Exploring efficient few-shot adaptation for vision transformers. TMLR 2022.
> [2] Basu S, Massiceti D, Hu S X, et al. Strong Baselines for Parameter Efficient Few-Shot Fine-tuning. arXiv 2023.*
>
> Our answer:
>
> Kindly note that the experimental setups of the above referenced works [1] and [2] are ***significantly different*** from ours, so  it is unfair to compare [1,2] with our methods.
>
> Specifically, there are several major differences in setups.
>
> 1) Backbone  is  different. The methods in [1, 2] adopt vision transformer (ViT) while we use classical convolutional networks (i.e., ResNet).
>
> 2) Pre-training is different. [1,2] adopts self-supervised learning (SSL) based on DINO (Caro et al, ICCV 2021), while we use traditional supervised learning.
>
> 3) Augmentations are different. [1,2] uses stronger data augmentations such as color jittering, Gaussian blur, solarization and multi-crop; in contrast, we use standard augmentations including color jittering, random horizontal clip and resized crop.
>
> In addition, [2] does not follow standard splits on Meta-Dataset, i.e., pretraining is performed on the whole ImageNet.
>
> Comparison of experimental setup
>
> |Method|Reference \[1\]|Reference \[2\]|GEMD-T|
> |:-|:-|:-|:-|
> |Backbone|ViT-Tiny & ViT-Small|ViT-Small|ResNet-18|
> |Pre-training|DINO-based SSL|DINO-based SSL|Supervised learning |
> |Augmentation|strong|strong|standard|

---

> > ### Comment · Reviewer_epv6 · 2023-11-22
> >
> > Thank the authors for the comprehensive feedback. I have no other questions.

---

> > > ### Author Response · Authors · 2023-11-22
> > >
> > > Thank you for your positive feedback. We are pleased to know that you increase the rating to 6 and appreciate your support for our work

---

### Comment · Area_Chair_s4NS · 2023-11-10
**Authors-Reviewers discussion starts today, ends on Nov 22**

Dear authors and reviewers,

@Authors: please make sure you make the most of this phase, as you have the opportunity to clarify any misunderstanding from reviewers on your work. Please write rebuttals to reviews where appropriate, and the earlier the better as the current phase ends on Nov 22, so you might want to leave a few days to reviewers to acknowledge your rebuttal. After this date, you will no longer be able to engage with reviewers. I will lead a discussion with reviewers to reach a consensus decision and make a recommendation for your submission.
IMPORTANT: your paper is lacking one or more reviews -- we are working very hard to solve this by contacting reliable emergency reviewers. Please check this page daily as new reviews are likely to appear.

@Reviewers: please make sure you read other reviews, and the authors' rebuttals when they write one. Please update your reviews where appropriate, and explain so to authors if you decide to change your score (positively or negatively). Please do your best to engage with authors during this critical phase of the reviewing process.

This phase ends on November 22nd.

Your AC

---

### Meta-Review · Area_Chair_s4NS · 2023-12-05

**Metareview:**

This meta-review is a reflection of the reviews, rebuttals, discussions with reviewers and/or authors, and calibration with my senior area chair. This paper is about few-shot learning and introduces two interesting new methods, with theoretical and experimental support. Several reviewers missed important points of the paper, and unfortunately not all reviewers have engaged with the rebuttals. In my opinion a significant revision and proper peer review seem unavoidable to really help this work shine at a further conference. I appreciate this will come as a disappointment to the authors: ICLR is a highly competitive venue and unfortunately the clarity of this submission is below the bar.

**Justification For Why Not Higher Score:**

A number of shortcomings in the paper and lack of enthusiasm from reviewers.

**Justification For Why Not Lower Score:**

N/A

---

### Decision · Program_Chairs · 2024-01-16

Reject